# CONCN: A high-resolution, integrated surface water-groundwater ParFlow modeling platform of continental China

**Chen Yang[1], Zitong Jia[2], Wenjie Xu[3], Zhongwang Wei[1], Xiaolang Zhang[4], Yiguang Zou[5],**

**Jeffrey McDonnell[6,7,8], Laura Condon[9], Yongjiu Dai[1], Reed Maxwell[10]**

[1]School of Atmospheric Sciences, Sun Yat-sen University, Zhuhai, China

[2]College of Water Sciences, Beijing Normal University, Beijing, China

[3]Institute of Geological Survey, China University of Geosciences, Wuhan, China

[4]Department of Geosciences, Florida Atlantic University, Boca Raton, USA

[5]Department of Geography, National University of Singapore, Singapore, Singapore

[6]School of Environment and Sustainability, Global Institute for Water Security, University of Saskatchewan, Saskatoon, Canada

[7]School of Geography, Earth & Environmental Sciences, University of Birmingham, Birmingham, UK

[8]North China University of Water Resources and Electric Power, Zhengzhou, China

[9]Department of Hydrology and Atmospheric Sciences, University of Arizona, Tucson, USA

[10]Department of Civil and Environmental Engineering, High Meadows Environmental Institute, Integrated GroundWater Modeling Center, Princeton University, Princeton, USA

*Correspondence to:*

*Chen Yang (yangch329@mail.sysu.edu.cn)*

*Yiguang Zou (e1144054@u.nus.edu)*

*Reed Maxwell (reedmaxwell@princeton.edu)*

**Abstract.** Large-scale hydrologic modeling at national scale is an increasing important effort worldwide to tackle ecohydrologic issues induced by global water scarcity. In this study, a surface water-groundwater integrated hydrologic modeling platform was built using ParFlow, covering the entire continental China with a resolution of 30 arcsec. This model, CONCN 1.0, has a full treatment of 3D variably saturated groundwater by solving Richards' equation, along with the shallow water equation at the ground surface. The performance of CONCN 1.0 was rigorously evaluated using both global data products and observations. RSR values show good to excellent performance in streamflow, yet the streamflow is lower in the Endorheic, Hai, and Liao Rivers due to uncertainties in potential recharge. RSR values also indicate good performance in water table depth of the CONCN model. This is an intermediate performance compared to two global groundwater models, highlighting the uncertainties that persist in current large-scale groundwater modeling. Our modeling work is also a comprehensive evaluation of the current workflow for continental-scale hydrologic modeling using ParFlow and could be a good starting point for the modeling in other regions worldwide, even when using different modeling systems. More specifically, the vast arid and semi-arid regions in China with substantial sinks (i.e., the end points of endorheic rivers) and the large uncertainties in potential recharge pose challenges for the numerical solution and model performance, respectively. Incompatibilities between data and model, such as the mismatch of spatial resolutions between model and products and the shorter, less frequent observation records, require further refinement of the workflow to enable fast modeling. This work not only establishes the first integrated hydrologic modeling platform in China for efficient water resources management, but it will also benefit the improvement of next generation models worldwide.

## 1. Introduction

China has been facing a persistent water crisis due to rapid socio-economic development and population growth (Jiang, 2009), resulting in the second lowest per inhabitant water supply among all countries worldwide (Pietz, 2017). The increasing water demand in China has been further exacerbated by more frequent hydrologic extremes, such as droughts and floods, driven by climate change and human activities. Water availability in China not only affects the nation's development trajectory but also influences the global food and supply chain (Collins and Reddy, 2022). Therefore, it is pressing to develop a consistent hydrologic modeling platform at national scale for water resources management, water quality control, and decision-making. Some work has begun in this regard. A national-scale groundwater model with a 10 km resolution based on MODFLOW has been built (Lancia et al., 2022), and national-wide natural streamflow was reconstructed using the Variable Infiltration Capacity (VIC) model with a 0.25° resolution (Miao et al., 2022). Additionally, regional groundwater models or hydrologic models with a groundwater component have been developed for focus areas, such as the North China Plain (Cao et al., 2013; Yang et al., 2020; Yang et al., 2023a), the Heihe River Basin (Hu et al., 2016; Tian et al., 2015), the Pearl River Basin (Wang et al., 2023; Yu et al., 2022), and the Jianghan Plain in the central Yangtze River (Jiang et al., 2022). These advances in China's modeling community are valuable for quantifying the fluxes, storage, and quality of streamflow and groundwater, thereby supporting the sustainable development of the country.

There is an increasing number of national and global modeling platforms worldwide for surface water, groundwater, or a combination of both. National-scale models include the US NOAA National Water Model (NWM) (Cosgrove et al., 2024), the USGS National Hydrologic Model (NHM) (Regan et al., 2019), the ParFlow (Parallel Flow) CONUS modeling platform (Maxwell et al., 2015; Yang et al., 2023b), the Canada National Water Model (Canada1Water) (Chen et al., 2020), the British Groundwater Model (BGWM) (Bianchi et al., 2024), and the national-scale models from Germany (Belleflamme et al., 2023; Hellwig et al., 2020), France (Vergnes et al., 2023), Denmark (Henriksen et al., 2003), Netherland (Delsman et al., 2023), and New Zealand (Westerhoff et al., 2018). Global models include the hydrologic model WaterGap and its groundwater component $G^3M$ (Reinecke et al., 2019; Müller Schmied et al., 2021), the hydrologic model PCR-GLOBWB and its associated groundwater models (Sutanudjaja et al., 2018; Verkaik

et al., 2024; De Graaf et al., 2015; De Graaf et al., 2017; Hoch et al., 2023), and Fan's global
groundwater model (Fan et al., 2013; Fan et al., 2017).
How to build a large-scale hydrologic model that balances high-performance with the trade-
off between resolution and computational efficiency is a critical issue in the hydrologic modeling
community, especially in groundwater modeling or modeling with a full treatment of groundwater.
However, it remains an open question since the subsurface is largely unseen. Reinecke et al. (2020)
compared the performance of several popular global groundwater models in New Zealand, along
with the New Zealand national groundwater model (Westerhoff et al., 2018). Reinecke et al. (2020)
attributed the departure of simulations from observations to model resolution, but Yang et al.
(2023b) suggested that the model's structure and parameters also play a role. Significant progresses
have been achieved in community discussions regarding model parameterization, evaluation,
calibration, and intercomparison (Gleeson et al., 2021; Condon et al., 2021; O'neill et al., 2021;
Tijerina et al., 2021). Yet, building a large-scale, high-resolution hydrologic model with satisfied
performance remains a challenging task (Reinecke et al., 2024; Devitt et al., 2021).
The most recent ParFlow CONUS 2.0 (Yang et al., 2023b) surface water-groundwater
integrated hydrologic model demonstrates excellent performances in both streamflow and water
table depth when compared with substantial observations collected from the US Geological Survey
(USGS) and other sources. However, the feasibility of its modeling workflow in other regions in
the world has not yet been evaluated. Here, we use the CONUS 2.0 workflow as a starting point
to build the modeling platform of continental China (CONCN). China has contrasting climatic
conditions, including large arid and semi-arid areas in the northwest with annual potential
evapotranspiration up to ~1400 mm (Li et al., 2014) and extremely wet condition in southeast with
annual precipitation exceeding 2000 mm (Han et al., 2023). The landforms are diverse,
encompassing snowpacks, wetlands, deserts, and plains. The topographic relief is dramatic,
ranging from the world's highest mountain ranges in Tibet to the sea level in coastal plains. All
these factors make China a favorable testbed for the CONUS 2.0 workflow, yet they also introduce
new challenges in the modeling. Additionally, US has databases of meteorology, hydrology,
topography, soil, and geology, along with relatively mature systems of data management and
sharing. In contrast, the existence and accuracy of some necessary data in China remain uncertain.
These differences challenge the transferability of the CONUS 2.0 workflow, necessitating
modifications during the CONCN modeling process. Hence, building the CONCN model is not
only essential for achieving national-scale consistent management of water resources but also
important for identifying the strengths and limitations of the workflow. This will help improve the
performance of next generation models at larger or global scales.
In the following sections, we first introduced the structure and parameters of CONCN 1.0,
including the construction of hydrologically consistent topography, hydrostratigraphy, and
potential recharge, which are the key components of the ParFlow model. We highlighted the
challenges in building the CONCN 1.0 model and described the strategies to overcome these
obstacles. We then evaluated the performances of the CONCN model in streamflow and water
table depth by both global data products and observations. The comparisons of CONCN model
with other model products are not intended to determine which model is better but rather to identify
the common problems faced by the modeling community. At the end of the paper, we also
discussed the challenges and opportunities in integrated hydrologic modeling for communities in
China.
Note that all performance evaluations in this paper are based on the RSR value which is the
ratio of the root mean squared error to the standard deviation of observations. An RSR value of 1.0
suggests a good performance while 0.5 suggests an excellent performance (O'neill et al., 2021).
However, the performances defined here are only for the comparison in this study, indicating the
capabilities of the model relative to the benchmark we used (i.e., global products or observations).
RSR values for different variables in this study (i.e., drainage area, streamflow, water table depth)
are not comparable. RSR values are generally not comparable with those in other case studies
using different models. Even for the same models used in this study, different observations and
different simulation periods represent different benchmarks and different system dynamics,
respectively, so it is hard to say the same RSR value represents the same performance of a model
(N. Moriasi et al., 2007; Schaefli and Gupta, 2007; Knoben et al., 2019).

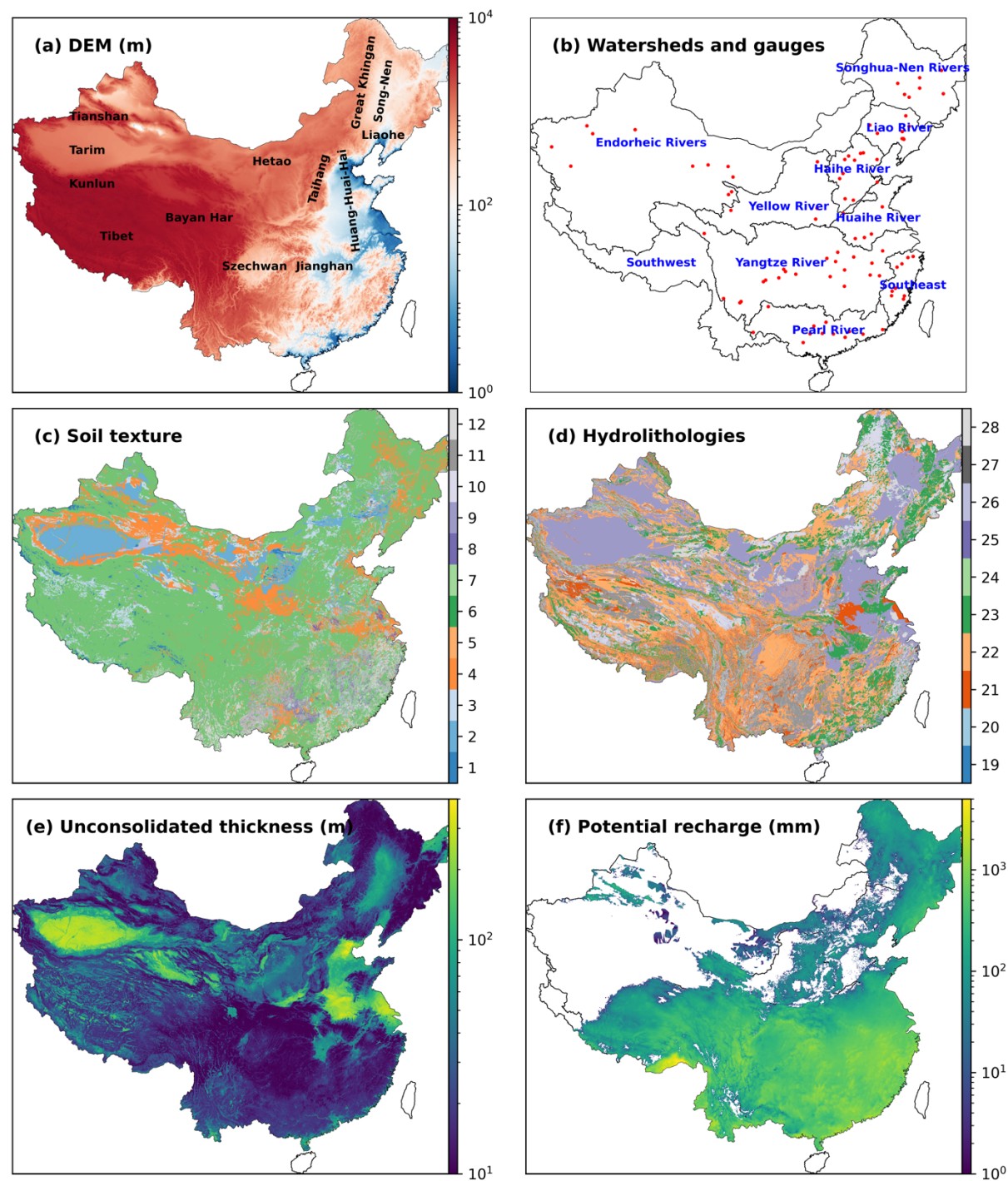

Figure 1. DEM processed by PriorityFlow and labeled with major basins, plains, and mountain ranges (a), major watersheds and streamflow gauges (red points) (b), soil texture of the top soil layer (the first layer from top to bottom) (c), hydrolithologies of the top layer (the fifth layer from top to bottom) (d), unconsolidated thickness (e), and annual potential recharge (f). The empty areas in (f) have potential recharge of zero in the model. Indictors of soil texture: 1. Sand, 2. Loamy sand, 3. Sandy loam, 4. Silt loam, 5. Silt, 6. Loam, 7. Sandy clay loam, 8. Silty clay loam, 9. Clay loam, 10. Sandy clay, 11. Silty clay, 12. Clay. Indicators of hydrolithologies: 19. Bedrock 1, 20. Bedrock 2, 21. f.g. sil. sedimentary, 22. sil. sedimentary, 23.

crystalline, 24. f.g. unconsolidated, 25. unconsolidated, 26. c.g. sil sedimentary, 27. carbonate, 28. c.g. unconsolidated. f.g., sil. and c.g. represent fine-grained, siliciclastic sedimentary, and coarse-grained, respectively.

## 2. Model parameterizations

The CONCN 1.0 model covers the entire continental China (Figure 1a) with a horizontal resolution of 30 arcsec (~1 km at the equator). Vertically, the CONCN model is composed of 10 layers with thicknesses of 300, 100, 50, 25, 10, 5, 1, 0.6, 0.3, 0.1 m from bottom to top. This structure results in 4865 and 3927 grid-cells in $x$ and $y$ directions, respectively, and a total of 98.8 million active grid-cells. Although we used the CONUS 2.0 workflow as a starting point for CONCN 1.0, modifications to the workflow were necessary, as mentioned in the introduction. One reason is primarily due to the data availability in China. This does not mean that the relevant data is completely missing, but rather that the data is not readily available for modeling purpose, or that its quality is uncertain. Another reason is due to the scientific progress that has occurred since the development of the CONUS 2.0 model. For example, the total model depth of CONCN 1.0 (492 m) is deeper than the depth of CONUS 2.0 (392 m). The increased model depth better closes the terrestrial hydrologic cycle, as groundwater contributes to global streamflow to a depth of ~500 m (Ferguson et al., 2023). The details of these modifications are discussed in the following sections.

### 2.1. Topographic processing

The most important two components of a ParFlow model are the topographic inputs and the hydrostratigraphy, which largely determine the model's performances of streamflow and groundwater, respectively. Since this is a surface water-groundwater integrated hydrologic model, topographic inputs may also influence the potential recharge to groundwater while hydrostratigraphy is crucial for accurate simulations of baseflow. Topographic inputs refer to slopes in the $x$ and $y$ directions, which are calculated from a digital elevation model (DEM) (Figure 1a). This DEM has been processed to ensure the D4 connectivity of the drainage network. D4 connectivity means that, within each grid-cell, streamflow is only allowed in east-west and north-south directions, but not allowed in diagonal directions. The original DEM used in this study is a data product with a resolution of 30 arcsec (Eilander et al., 2021), which was upscaled from the MERIT Hydro DEM with a resolution of 3 arcsec (~90 m at the equator) (Yamazaki et al., 2019),

using an Iterative Hydrographic Upscaling approach (hereafter abbreviated as IHU DEM). The DEM was processed using PriorityFlow, which was developed during the CONUS 2.0 modeling (Condon and Maxwell, 2019). Note that the horizontal resolution of the CONCN 1.0 model (i.e., 30 arcsec) is set to be consistent with the resolution of this IHU DEM.

Reference stream networks are preferred as inputs in PriorityFlow to improve the drainage performance. The challenge is that we do not have a consistent gridded stream network at the national scale with a resolution close to that of CONCN 1.0, whereas a network with 250 m resolution from the National Water Model (NWM) is available for CONUS 2.0 (Zhang et al., 2021). As a replacement, we generated stream networks from the IHU flow direction of D8 connectivity. Then we checked the generated networks with the vector networks generated from the 3 arcsec MERIT Hydro flow direction (Lin et al., 2019). The initial threshold of the drainage area used to generate the input networks from the IHU flow direction was set to 300 $km^2$. During the processing using PriorityFlow, we refined some input networks locally by gradually decreasing the threshold. Such refinements are necessary in areas with flat topographies (e.g., the Huang-Huai-Hai plains and coastal plains in Figure 1a), where flow directions are difficult to identify without additional reference networks. Endorheic rivers are common in Northern and Northwest China. Sinks, the end points of these endorheic rivers, are also important to constrain flow directions and thus to generate accurate D4 stream networks. Manual refinements of input networks, including the sinks, were iterative processes until the networks generated by PriorityFlow appeared consistent with the vector networks and there were no obvious ponding cells in runoff simulations. A total of 924 sinks were identified in CONCN 1.0, compared to only 131 sinks in CONUS 2.0, which increases the difficulty of the numerical solution, as ParFlow currently does not handle such water bodies.

In addition to the qualitative evaluation described above, we also compared the drainage areas generated by PriorityFlow with those in IHU and with 294 observations collected from the literature (Yin et al., 2024). Increasing performance was observed during the iterative processing and the final performances are shown in Figure 2. The PriorityFlow and IHU drainage areas match well, with an RSR value smaller than 1, indicating a good performance (Figure 2a). An additional interesting finding is the scaling relationship between drainage areas and frequencies. The comparison with observations shows excellent performance, as the RSR value is smaller than 0.5 (Figure 2b). Deviations from the 1:1 line were observed for drainage areas smaller than 100 $km^2$, as we focused more on drainage areas larger than 100 $km^2$ during the processing.

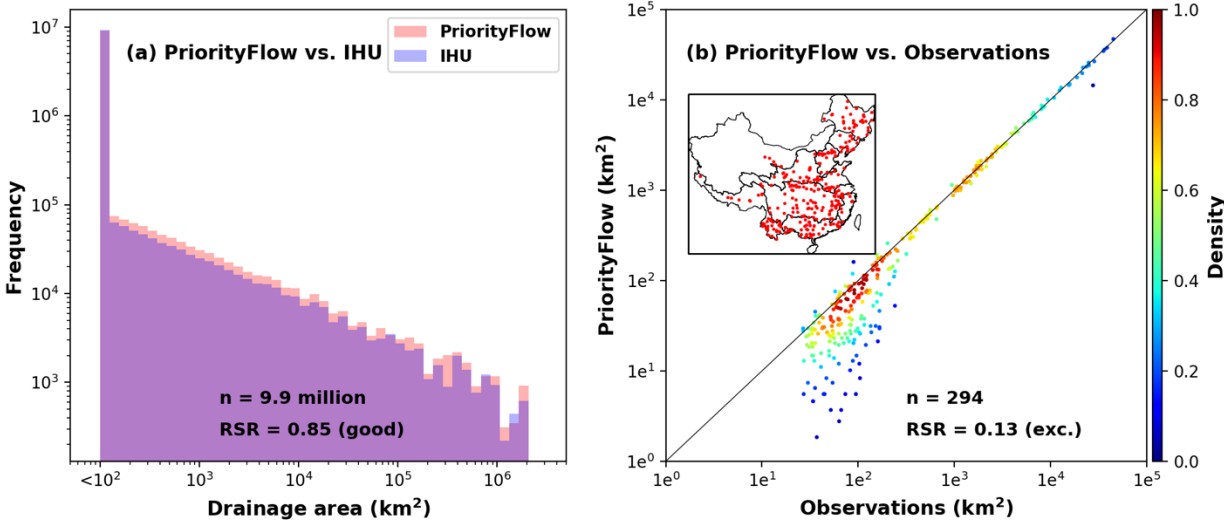

**Figure 2. Evaluating the drainage performance of the topography processed by PriorityFlow using (a) IHU drainage areas and (b) observations collected from literature.**

## 2.2. Hydrostratigraphy

The general structure of the hydrostratigraphy is composed of shallow soils and deeper hydrolithologies. The latter includes both unconsolidated and consolidated sediments (Fan et al., 2007; Huscroft et al., 2018). The details of the implementation are as follows: the top 2 m consists of four soil layers (0.1, 0.3, 0.6 and 1.0 m from top to bottom). The relative percentages of sand, clay, and silt in each layer were derived from a global dataset of soil hydraulic properties (Dai et al., 2019) with a 30 arcsec resolution. Twelve soil textures (Figure 1c) were then built from these percentages, based on the soil classification defined by the US Department of Agriculture. Hydrolithologic categories (Figure 1d) were reclassified from the permeabilities of GLHYMPS 1.0 (Gleeson et al., 2014), which was built by categorizing lithologies in the global lithology map, GLiM. GLiM was compiled by using the geologic map of 1:2.5 million scale in China area released by the China Geological Survey in 2001 (Hartmann and Moosdorf, 2012). Then *e*-folding, representing variations of hydraulic conductivity with depth and terrain slope, was applied to each of the deep six layers (Fan et al., 2007; Tijerina-Kreuzer et al., 2023). Flow barriers (Figure 1e) were implemented at the interfaces between unconsolidated and consolidated sediments via multiplying the hydraulic conductivities by 0.001 to represent a potential confining layer (De graaf et al., 2020; Huscroft et al., 2018). This concept represents the lumped effects of low-permeability sedimentary materials in the unconsolidated layer. The dataset we used to represent the interface

depths was specifically developed for China (Yan et al., 2020) and is more accurate than the global
version used in CONUS 2.0 (Shangguan et al., 2017).
We adopted this hydrostratigraphy as it is the most convincing scheme from CONUS 2.0,
selected through rigorous hydrologic modeling tests from hundreds of combinations of different
components, such as the distribution of hydrolithologic categories, anisotropy of some categories,
implementation of confining layers, *e*-folding of the hydraulic conductivities, total model depth,
and constant or variable depths of confining layers (i.e., flow barriers) (Swilley et al., 2023;
Tijerina-Kreuzer et al., 2023). The hydraulic parameters for each soil texture and hydrolithologic
category (e.g., hydraulic conductivity, porosity, specific yield, and parameters of the van
Genuchten model) were adopted from Schaap and Leij (1998) and Gleeson et al. (2014), with
slight calibration in the CONUS models (Maxwell et al., 2015; Yang et al., 2023b). The parameter
configuration assumes that each soil texture or hydrolithologic category has a set of representative,
scale-independent hydraulic parameters.
**2.3. Potential recharge**
The construction of potential recharge used to drive the model is the most challenging part in
this modeling work. Potential recharge here refers to the multi-year averaged precipitation (P)
minus evapotranspiration (ET), i.e., P-ET. Uncertainties of such hydrometeorological variables are
always high. For example, the relative standard deviation (standard deviation relative to the mean)
of the annual mean ET from 12 global products using different approaches reaches 50% (Jiménez
et al., 2011). Given this issue, the P and ET datasets selected for CONUS 2.0 were generated from
a VIC modeling framework (Livneh et al., 2015), which adjusts P for orographic effects and
ensures closure of the land surface water budget. Therefore, uncertainties of all hydrologic
variables were constrained within a consistent modeling system. However, datasets of P and ET in
China generated by various approaches have inconsistent uncertainties, and a closed water balance
for all hydrologic components is absent. Uncertainties in P-ET may further accumulate during data
processing (e.g., resampling, interpolations, and transforms) due to differences in the
spatiotemporal resolutions of the P and ET products and the CONCN model. Additionally, the
record lengths and the data quality of some datasets are hard to balance, also challenging the
accurate representation of a long-term average state of the predevelopment condition. We collected
four precipitation products and five ET products generated based on (1) interpolation of the
measurements (Han et al., 2023), (2) models including the Penman-Monteith equation (Running
et al., 2021), the complementary relationship model (Ma et al., 2019), and the land surface model
(Muñoz-Sabater et al., 2021), and (3) model-data fusion (Huang et al., 2014; Peng, 2020; Niu et
al., 2020; Zhang et al., 2019).
An accurate evaluation of different products was not conducted, as it is beyond the scope of
this study. More importantly, it will take time for the community to gradually improve the quality
of these datasets. We roughly evaluated the products using prior knowledge of some focus areas.
For example, we randomly selected several locations and compared the multi-year average levels
of P or ET with the commonly known levels. We used the same approach to evaluate the P-ET
generated by combining different P and ET datasets. For example, P-ET showed negative values
in some arid and semi-arid regions in northwest China where P-ET should be a dominant source
for known rivers. Although ERA5-Land products also provide P and ET datasets under a consistent
modeling framework with high enough resolution (~9 km at the equator), its precipitation dataset
is obviously lower than that constructed using interpolation of substantial measurements in Han et
al. (2023). The best combination of P (Han et al., 2023) and ET (Niu et al., 2020) in the evaluations
was selected to create the average state of potential recharge from 1981 to 2010 (Figure 1f).
However, errors induced by uncertainties from P and ET, especially ET, are still evident in some
regions, such as the Tarim River Basin, the Heihe River basin, and the Haihe River Basin (i.e., the
North China Plain). The inaccuracy estimation of potential recharge would affect the simulated
groundwater and streamflow as discussed in the following sections.
**2.4. Manning's roughness coefficients**
The Moderate Resolution Imaging Spectroradiometer (MODIS) Land Cover Type (MCD12Q1)
version 6.1 data product with a 500 m resolution (Friedl and Sulla-Menashe, 2022) was used to
build the distribution of Manning's roughness coefficients, which are necessary for calculating
streamflow, and will also be required by the Common Land Model (CLM) (Dai et al., 2003) in the
future transient ParFlow-CLM model (Kollet and Maxwell, 2008). The land cover types in this
product follow the International Geosphere-Biosphere Programme (IGBP) classification, which is
consistent with the classification required by ParFlow-CLM. In the modeling of CONUS 2.0, a
land cover map with a higher resolution of 30 m was reclassified into the IGBP classification.
Some products with resolutions higher than 500 m are also available in China (Yang and Huang,
2021), but their coarse classifications prevented us from reclassifying the types to subtypes. Stream
networks were generated using PriorityFlow with a threshold drainage area of 50 km$^2$, and stream
orders were calculated based on the Strahler stream order (Strahler, 1957). Manning's roughness
coefficients were set to vary by land cover type and were further adjusted in stream channels,
decreasing in value with increasing stream order. The values of Manning's roughness coefficient
for each land cover type and each stream order were adapted from the National Water Model
(Gochis et al., 2015) and a previous study (Foster et al., 2020).

## 3. ParFlow modeling platform

ParFlow simulates the movement of 3D variably saturated groundwater and 2D surface water
simultaneously by solving Richards' equation with the shallow water equation as the top boundary
(Kollet and Maxwell, 2006). CONCN 1.0 uses a terrain following grid, which significantly reduces
the computational load compared to an orthogonal grid (Maxwell, 2013). The model was
initialized with a uniform water table depth (WTD) of 2 m and was driven by the average potential
recharge of 1981–2010 to achieve a quasi-steady state for evaluations in following sections. All
faces of the model, except the top boundary, are no flow boundaries. We ran the model using the
seepage face boundary condition on top of the model until the total storage change was less than
1% of the potential recharge. This is to form the topography-driven patterns of water table.
Afterward, the overland kinematic boundary condition was enabled to generate river systems. The
spinup continued until the total storage change was less than 3% of the potential recharge. River
systems quickly reached a quasi-steady state in groundwater convergence areas, which had already
been identified in the first stage. This two-phase spinup process omitted unnecessary surface water-
groundwater interactions during the early stage to improve computational efficiency. Although the
dimension of CONCN 1.0 is comparable to CONUS 2.0, CONCN 1.0 required more time for
spinup because rivers in arid and semi-arid regions take longer to reach a quasi-steady state, as the
water is limitedly recharged by local precipitation but is sourced from the far away upstream.
The Newton-Krylov approach is employed to solve this large nonlinear system, which is
discretized on a finite difference grid in an implicit manner. Parallel scalability of the model is
ensured by using a multi-grid preconditioner. Thresholds of nonlinear and linear iterations are 1e$^{-5}$
and 1e$^{-10}$, respectively, to ensure proper convergence. The model was run on Princeton Della
GPU cluster using four 80-GB NVIDIA A100 GPU cards, or on the NCAR Derecho
supercomputer using 4096 processor cores across 32 nodes. Each node on Derecho is equipped
with 3rd Gen AMD EPYC™ 7763 Milan CPUs.
**323 Figure 3. Simulated streamflow and water table depth by CONCN 1.0, representing the**
**324 average state from 1981 to 2010.**


## 4. Simulations and evaluations

The simulated streamflow and WTD are shown in Figure 3. Patterns of streamflow (Figure 3a) reveal a contrast between wet and dry regions, generally consistent with the monsoon and non-monsoon regions. Large river systems in the monsoon region are well represented, such as the Yellow River in northern China, the Yangtze River and the Pearl River in southern China, and the Songhua, Nen, and Liao Rivers in northeast China. During the spinup, we observed that the Yellow River is primarily recharged by water sourced from the Bayan Har Mountain ranges and by a small amount of local groundwater. The number of river segments recharged by precipitation increases downstream the Hetao Plain. River systems in northwest China are also visible, though future work is needed to improve accuracy by reducing uncertainties in potential recharge. The WTD (Figure 3b) presents topography-driven patterns, showing shallow water tables in the Huang-Huai-Hai Plain, the Jianghan Plain, the Liaohe Basin, and the Songnen Plain. The water table is also shallow inside the Tarim Basin, where the terrain is flat, even though annual precipitation there is lower than 50 mm. Deep water tables are distributed along the Tianshan and Kunlun Mountain ranges, the Taihang-Great Khingan Mountain ranges, and the transition area from the Tibet Plateau to the Szechwan Basin.

The performance of CONCN 1.0 was comprehensively evaluated by both data products and observations. In the evaluation using measured observations, it is difficult to ensure that the duration of the records is consistent with that of the potential recharge (1981–2010), as streamflow or groundwater observations earlier than 2000 are hard to collect. This mismatch between the simulation and observation periods may cause discrepancies between simulated and observed values, due to the different drivers resulting from interannual variations of P and ET. This highlights a new challenge relative to CONUS 2.0 modeling, as publicly accessible observations in the US date back to 1900 or even earlier.

### 4.1. Evaluation of streamflow

We compared the simulations of CONCN 1.0 with a global streamflow dataset, GRADES-HydroDL (Yang et al., 2023c). The daily streamflow from 1980 to present is estimated for 2.94 million river reaches by applying a Long Short-Term Memory (LSTM) model on a 0.25° grid, developed following Feng et al. (2020), and then coupling the LSTM model with a river routing model (RAPID) (David et al., 2011). River reaches with drainage areas larger than 1000 km$^2$ were

selected, and those with drainage areas larger than 120% or smaller than 80% of the PriorityFlow
drainage areas were further filtered out. For each of the selected 23,609 reaches, streamflow during
the potential recharge period (1981 to 2010) was averaged and compared with the simulation of
CONCN 1.0. Locations of the selected reaches and a scatterplot of simulations vs. GRADES-
HydroDL are shown in Figure 4. Overall, we see comparable performances of CONCN 1.0 and
GRADES-HydroDL, with an RSR value close to 1. Smaller streamflow values are more scattered
in the plot due to the uncertainties associated with smaller drainage areas.

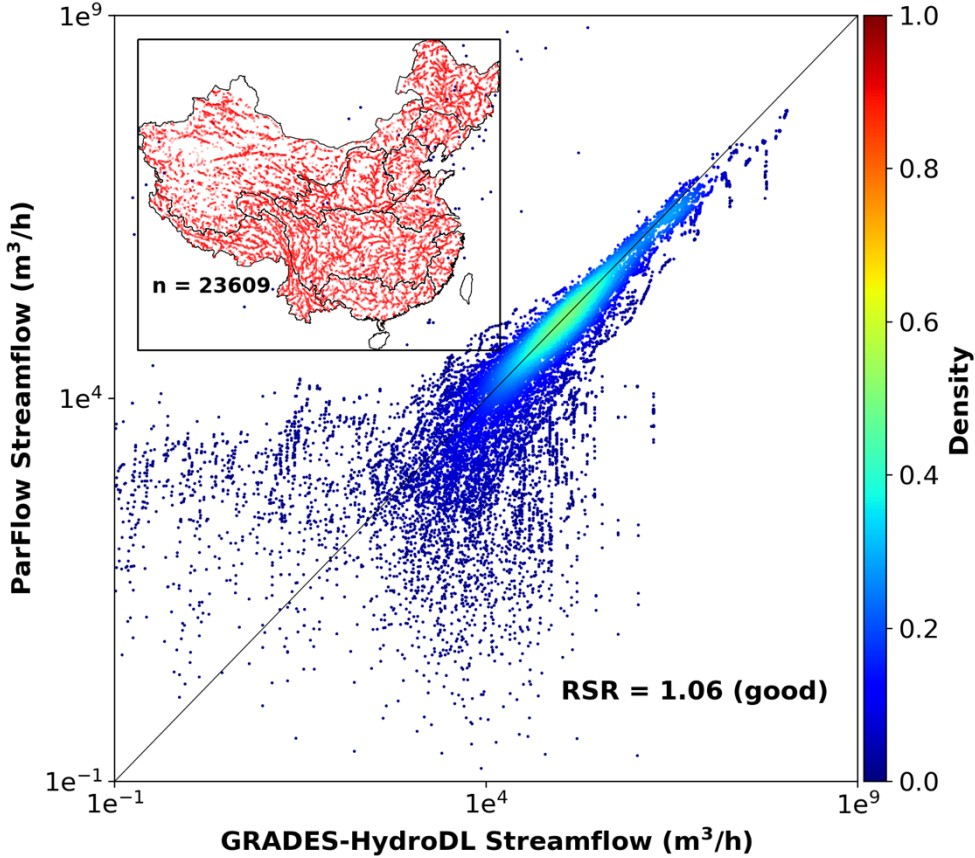

**Figure 4. Scatterplot of simulated streamflow vs. GRADES-HydroDL. Locations of the**
**selected reaches for comparison are shown in the upper left corner.**

We collected streamflow observations at 95 gauges from the annual River Sediment Bulletin
of China, with 88 gauges available for evaluation. Five gauges were removed because we could
not find their locations (i.e., latitude and longitude) in the lookup table of national gauges, one of
two very close gauges was also removed, and one gauge in Hainan province was excluded as it is
out of the modeling domain. Locations of the 88 gauges are shown in Figure 1b, covering most of
the modeling domain to ensure an impartial evaluation. However, the number of gauges is
obviously limited, and augmenting the database for this modeling platform will take time. The
observations include monthly records spanning from 2002 to 2021. Although most gauges do not
have a complete 20-year record, each gauge has at least a two-year record. Scatterplots of
simulations vs. observations are shown in Figure 5. Most basins show good to excellent
performances, with RSR values close to 1.0 or smaller than 0.5. Simulated streamflow of the
Endorheic, Haihe and part of the Liao Rivers is much lower than observed. This is likely due to
uncertainties in potential recharge, as discussed in section 2.3 and the fact that simulations at these
gauges are mainly baseflow sourced from groundwater. Slight deviations are also seen along the
mainstream of the Yangtze and Yellow Rivers, likely caused by hydraulic engineering, such as dam
operations.

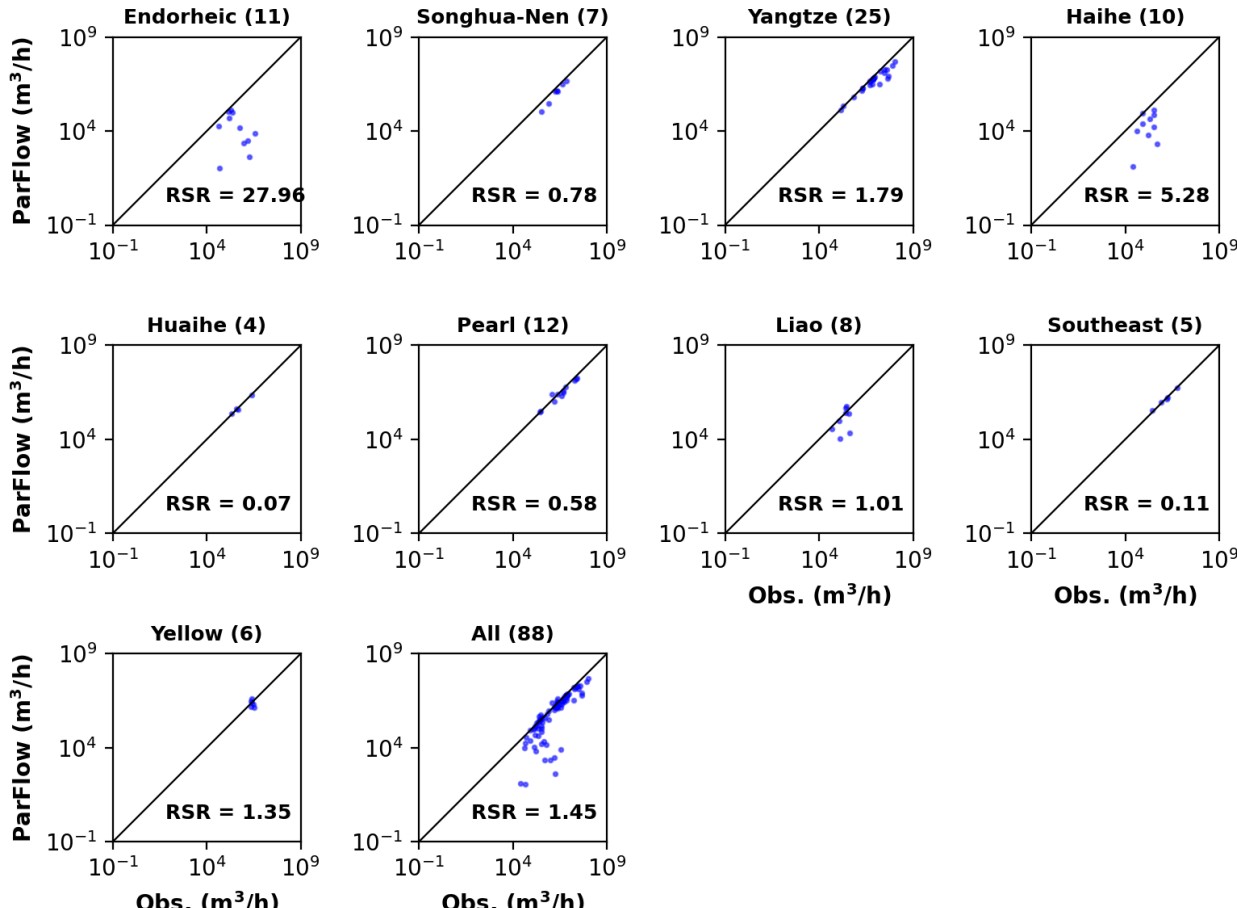

**Figure 5. Scatterplots of simulated vs. observed streamflow for basins in Figure 1b and the**
**entire CONCN 1.0 domain.**

## 4.2. Evaluation of water table depth

WTDs generated by two global groundwater models were collected (Figures 6a-b). Both models have a horizontal resolution of 30 arcsec, but their formulations and vertical structures differ significantly from CONCN 1.0. The first model is a horizontal, two-dimensional groundwater model of 40 layers. It describes 1D soil water movement within each column using Richards' equation and 2D lateral flow among columns using Darcy's law under the Dupuit-Forchheimer assumption. It is an inverse modeling originally developed by Fan et al. (2017) and later updated in 2020 [as eLetters in Fan et al. (2013)]. The water table in Figure 6a is the average of hourly dynamics from 2004 to 2014. The second model, GLOBGM v1.0, is a three-dimensional groundwater model with two layers, driven by outputs from PCR-GLOBWB (Verkaik et al., 2024). GLOBGM v1.0 is a steady-state model representing the average state for the period 1958–2015. It is a refined version of the 5 arcmin PCR-GLOBWB-MODFLOW model (De Graaf et al., 2015; De Graaf et al., 2017). Though GLOBGM v1.0 is not calibrated, its predecessor (De Graaf et al., 2017) was calibrated.

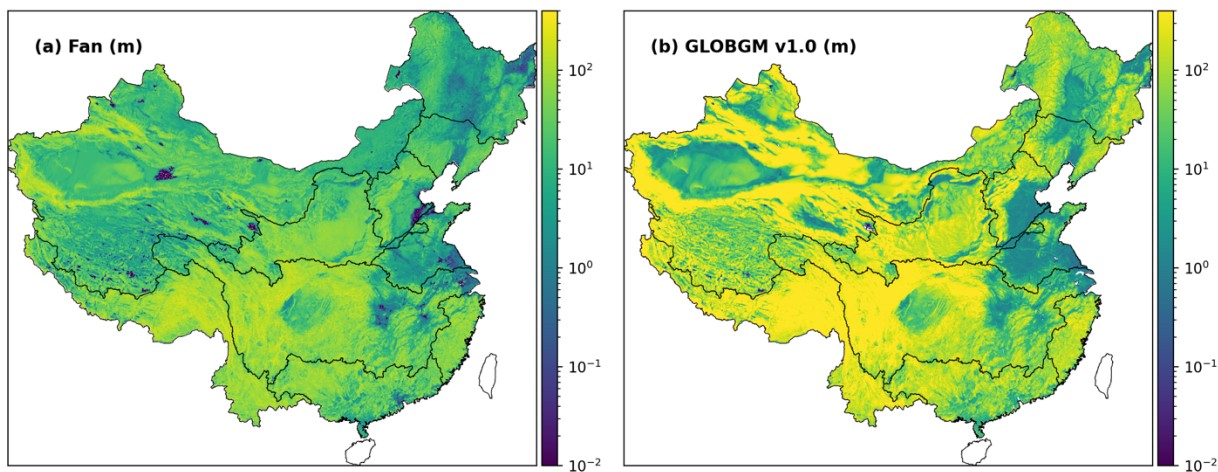

**Figure 6. Water table depths from two global models in CONCN 1.0 domain: (a) Fan's model and (b) GLOBGM v1.0.**

We collected monthly observations of hydraulic head in 8563 wells in 2018. After removing wells located out of the model domain, in confined aquifers, and on ParFlow river channels, 2436 wells remained for evaluation (Figure 7a). The annual means of WTD were calculated by subtracting hydraulic heads from well elevations measured at the land surface. These wells are part of the national groundwater monitoring network maintained by the Ministry of Land and Resources. We collected the data by digitizing the China Geological Environmental Monitoring

Groundwater Level Yearbook of 2018 and then double-checked the data to avoid errors. The yearbook, which started from 2005, has currently been updated to 2021. We fully understand that one-year monthly observations cannot represent the long-term average state of water table. An ongoing effort is being made to digitize all data in the yearbook and apply QA/QC (quality assurance and quality control) on the digitized data, although it will take a few years to finish.

We compared WTDs simulated by the three models with observations (Figure 8). RSR values show generally good performance for all three models. CONCN 1.0 has RSR values that fall between the two global models (0.88 compared to 0.80 and 1.41), with a bias towards shallow water tables. The residuals of WTD for each model are shown in Figures 7b-d. Each subplot also shows the decrease of groundwater storage based on GRACE data (Zhao et al., 2023), which is classified into three levels: moderate, rapid, and dramatic. The decrease of groundwater storage is mainly observed in Northern China, such as the Song-Liao Plains, the North China Plain, the Hetao Plain, and the northern edge of Tarim Basin. Agriculture, with intensive groundwater pumping for irrigation, is well-developed in these areas. While model simulations represent natural conditions without groundwater pumping, simulated water tables might be expected to be shallower than observed, i.e., negative WTD residuals, given anthropogenic impacts. However, the residuals from the Fan et al. model show positive residuals in these areas (Figure 7c). Similarly, positive residuals of GLOBGM v1.0 are found in Tibet and the Song-Liao Plains, where the decrease of groundwater storage is likely significant. Given the uncertainty in groundwater pumping and the lack of extraction data, it is challenging to represent these processes in large scale models. It is important to note that Fan's model is an inverse model and the predecessor of GLOBGM v1.0 is a calibrated model while the ParFlow CONCN 1.0 model is uncalibrated. Groundwater observations are sparse, many uncertainties exist, particularly in the subsurface architecture, and extraction from wells is unknown, creating a substantial modeling challenge. Calibration of large-scale groundwater models to observations becomes particularly challenging given that both extraction and hydraulic conductivity will lower water tables and can have an equal impact on RSR values.

438

**Figure 7. Well locations (a) and residuals of water table depth for each model (b-d). Residuals here refer to the differences between simulations and observations. The background shows the average decease of groundwater storage from 2003 to 2020 based on GRACE data (Zhao et al., 2023). The decrease is classified into three levels: moderate, rapid, and dramatic.**

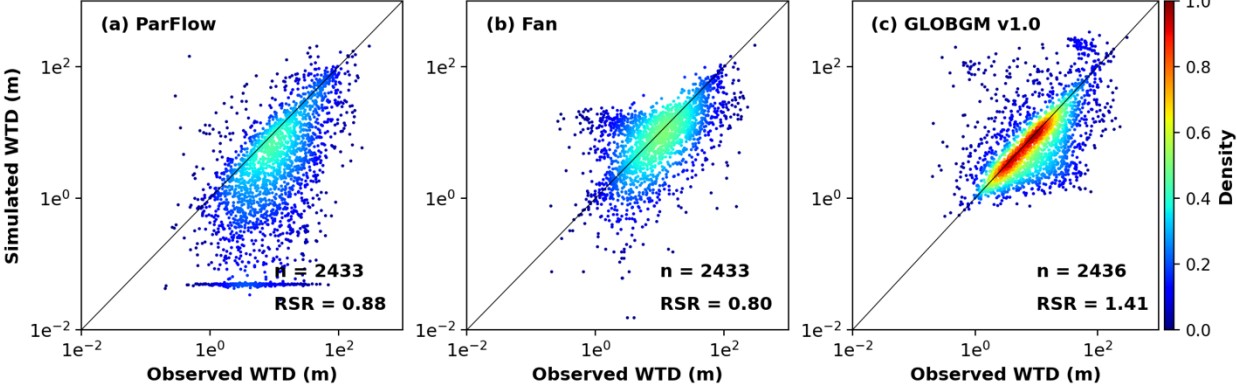

**Figure 8. Scatterplots of simulations vs. observations of water table depth. The simulations are long-term average water table depths of 1981–2010 in current work, 2004–2014 in Fan's model, and 1958–2015 in GLOBGM v1.0. The observations are the averages of 2018.**

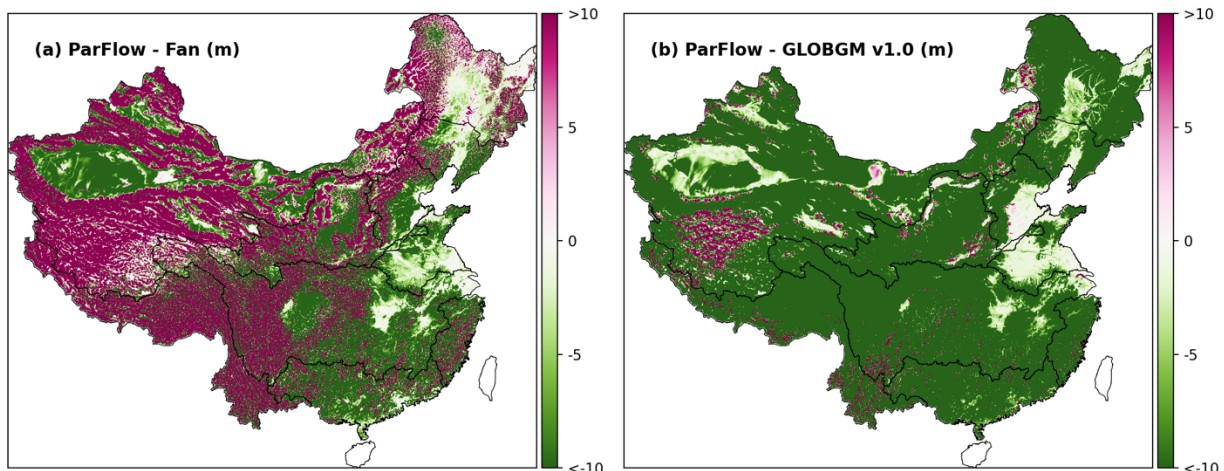

**Figure 9. Differences of water table depth between CONCN 1.0 and two global models: (a) the difference between CONCN 1.0 and Fan's model and (b) the difference between CONCN 1.0 and GLOBGM v1.0.**

## 5. Discussion

All three models show deep simulated water tables located just below the Szechwan Basin, at the boundaries of the Yangtze River and the Pearl River Basins (Figure 7). This finding may suggest a higher potential recharge in 2018 compared to the historic period (1981–2010), which was used to drive the model. Notably, these areas are part of the extensive Karst regions (Wang et al., 2019), where unique potential recharge and groundwater movement may occur along preferential flow pathways (e.g., fractures and conduits) (Hartmann et al., 2017). Given that ParFlow modeling has demonstrated acceptable performance in Karst regions in previous studies (Srivastava et al., 2014; Yang et al., 2023b), we didn't apply specific adjustments to model inputs for these regions. However, we assigned higher hydraulic conductivities in Karst regions, assuming that Karst aquifers behave similarly to porous media at an approximate 1 km scale (i.e., the model resolution). This assumption may simplify the Karst geology and we acknowledge its limitations, as the simulated deep water-tables could also result from the underlying Karst geology. More specifically, wells are easily created in places lacking prominent Karst features, where local hydraulic conductivities are relatively lower. However, the higher effective hydraulic conductivity of a grid cell may generate deep water tables without representing such subgrid heterogeneities.

In southern China where it is wet, CONCN 1.0 shows shallow water tables, whereas Fan's model and GLOBGW v1.0 predict relatively deeper water tables (Figures 7b-d). These differences likely arise from the distinct model formulations. ParFlow integrates overland flow and

groundwater movement by simultaneously solving Richards' and shallow water equations via
shared nodes in the top layer. As a result, WTDs in wells near rivers are likely underestimated, due
to the widened rivers in the model, resulted from the model's resolution of approximately 1 km.
WTDs in some wells located too close to rivers, e.g., tens or hundreds of meters which are smaller
than the model resolution, cannot be captured at all, as the grid cell has been fully saturated.
However, monitoring wells are typically located near rivers, which explains the shallow WTDs
generated by CONCN 1.0 in southern China (Figure 7b). Fan's model and GLOBGW v1.0, which
account for river-groundwater interactions, uses the difference between groundwater head and
river level (Fan et al., 2017; De Graaf et al., 2017). In these two models, rivers and the top
subsurface layer are loosely coupled without shared pressure heads. In other words, rivers can
flexibly carve the topography and groundwater with levels lower than the land surface can
discharge to rivers. Additionally, WTDs, even in grid-cells with rivers, can be calibrated, which is
not possible in ParFlow due to the integrated formulation. Though Fan's model and GLOBGW
v1.0 use similar formulations for groundwater-river interactions, WTDs of Fan's model are
shallower than those of GLOBGW v1.0 in southern China, which highlights other uncertainties of
the two models (Reinecke et al., 2024).
To avoid the bias in evaluation caused by well locations as they are concentrated near rivers,
we plotted the differences of WTDs between CONCN 1.0 and the two global models in Figures
9a-b. Results indicate that Fan's model generally produces shallower WTDs, whereas GLOBGW
v1.0 simulates deeper WTDs, i.e., an intermediate performance of CONCN 1.0, which expands
the understandings in riparian areas to the entire modeling domain. The significant discrepancies
of WTDs across the three models highlight substantial uncertainties in WTDs simulated by current
large-scale groundwater models, which cannot be fully revealed using the limited available
observations. This underscores the need for further efforts in parameterizations and formulations
of the models in this modeling community. Reinecke et al. (2024) found that WTDs generated by
global models are strongly correlated with topography (i.e., slope) yet exhibit minimal climatic
influences. In contrast, WTDs of Fan's model and observations show weaker correlations with
topography and can be further differentiated in water-limited and energy-limited regions. We show
generally similar results in Figure 10, where red and blue are used to differentiate the Spearman
rank correlations and boxplots in wet and arid regions based on potential recharge in Figure 2f.

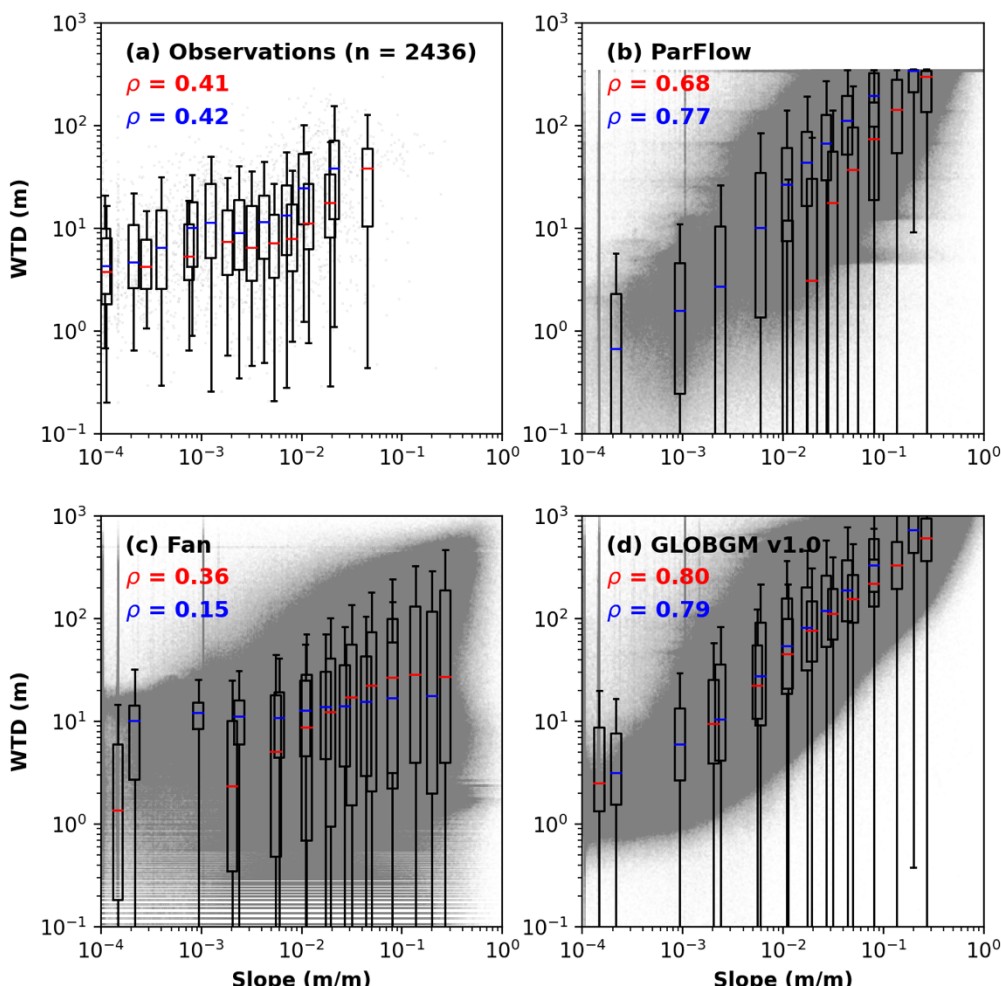

**Figure 10. Variations of water table depth with topographic slope for observations (a) and simulations of three models (b-c). Spearman rank correlations shown in each subplot are based on the point cloud. Boxplots (without outliers) for better visualization of the point cloud are created by evenly separating slopes to ten bins. Boxplot and Spearman rank correlations in each subplot are distinguished by potential recharge. Red and blue are for points located in regions with and without long-term average potential recharge (i.e., P-ET>0 and P-ET=0).**

In Figure 10, WTDs of CONCN 1.0 and GLOBGM v1.0 show strong correlations with topography, whereas observations and WTDs of Fan's model exhibit weaker correlations. The shallower WTDs simulated by ParFlow in wet regions, as discussed earlier, may decrease the correlation with topography, yet it is interesting to find that the terrain following grid of ParFlow results in slightly lower correlations than GLOBGM v1.0. Unlike the findings in Reinecke et al. (2024), correlations of observations with topography in different climate regions appear similar. This different finding could be attributed to the predominance of agricultural areas in the

observation dataset, where groundwater pumping likely exerts significant influence. The limited number of observations in this study is another potential limitation. In addition to the general explanation for the different correlations among models in Reinecke et al. (2024), the formulation of Fan's model may play an important role. By incorporating plant use of groundwater and dynamic root uptake depth in an inverse modeling based on inferred ET from remote sensing, Fan's model may introduce stronger regulations on water table via recharge, thereby diminishing the controls from (or the sensitivity to) topography. Additionally, Fan's model relies on remote sensing ET occurred under irrigation conditions, which includes the ET induced by irrigation. This additional ET is likely derived from groundwater, amplifying the effects of plant water use compared to natural conditions. This phenomenon may also explain the deeper water tables predicted by Fan's model in agricultural areas, as shown in Figure 7. These findings suggest that groundwater models should account for plant water use from deep soil or groundwater. Recharge estimated by hydrological or land-surface models with limited soil depths and/or without lateral groundwater convergence may be insufficient. Furthermore, uncertainties arising from human disturbances, such as groundwater pumping, should also be quantified.

## 6. Summary and forward outlook

In this study, we built the first surface water-groundwater integrated hydrologic modeling platform of the entire continental China with a high resolution using ParFlow. This CONCN 1.0 model was rigorously evaluated by both data products and observations, based on RSR values. Comparisons with observations show good to excellent performances in streamflow and water table depth. Comparisons with global data products show comparable performance of streamflow to the global model and an intermediate performance of water table depth among global models. These results also demonstrate the transferability of the modeling workflow using ParFlow. However, we also recognize the challenges inherent in large scale hydrologic modeling. Data quality and/or availability (e.g., for direct use or quick access) presents a significant challenge during modeling. The vast arid and semi-arid regions of China further increase uncertainties in input data, such as potential recharge. As a result, lower simulated streamflow is observed in northwest China and in the Haihe and Liao River Basins. Significant uncertainties in simulated water table depth are identified in current large-scale groundwater models, which might be attributed to the different parameterizations and formulations of the models, necessitating continuous efforts from the community.

We hope that our work can catalyze conversations and collaborations between various
communities involved in hydrologic modeling, geological surveys, model development, data
products, and data monitoring/sharing. Clearly, all efforts are aimed at improving the efficiencies
and capabilities of large-scale hydrologic modeling, which is powerful to address diverse
ecohydrologic issues and accelerate scientific discoveries across multiple disciplines. Below, we
summarize both the challenges and opportunities that require the attention and collaborative efforts
of the hydrology community and beyond.
(1) Human activities related to water resources are intensive in China, such as the long-term
groundwater pumping in Huang-Huai-Hai plains, the South-North water transfer projects,
the operation of the Three Gorges Dam, and the revegetation in the Loess Plateau. Flash
extremes are also becoming more frequent, such as the Yangtze drought (August 2022) and
the storms in Zhengzhou (July 2022) and Beijing (July 2023). These factors make China
one of the world's most significant ecohydrologic hotspots. Integrated hydrologic
modeling systems are essential to address these issues. While local and regional models
have been developed in recent years, modeling platforms with high resolution at larger or
national scales are still lacking, hindering efficient water resources management and timely
decision-making across multiple scales.
(2) Hydrologic processes, especially groundwater at the hillslope or catchment scales, play
important roles in terrestrial water and energy cycles, yet they are often oversimplified or
poorly represented in Earth system models. Many studies conducted in China on critical
hydrologic questions have focused on limited components of the hydrologic cycle.
Therefore, it is urgent to build large-scale hydrologic models and couple them with regional
weather or climatic models to better understand the terrestrial hydrologic cycle in China.
More importantly, the modeling should go beyond water balance to include flow paths or
water quality to gain a deeper understanding of the food-energy-water nexus and to conduct
risk assessment in the changing world.
(3) Large-scale hydrologic modeling relies on massive amounts of data for various input
variables. Discrete observations are often not user-friendly for direct use by modelers. Data
products help fill spatial and temporal gaps and are necessary for effective modeling. Many
of the products currently used only emerged in recent years, making large-scale hydrologic
modeling inefficient in China and other parts of the world in earlier times. The rapid
development of global data products suggests that now it is the ideal time to build large-
scale, consistent hydrologic modeling platforms. However, high-quality data products
within a consistent framework are still lacking and inter-evaluations between different
products could help constrain uncertainties from various sources (see section 2.3).
(4) We also need to leverage the strengths of local documents in China. The hydrolithologies
of GLHYMPS 1.0 were built using the global lithology map GLiM, which relies on the
geological map with a scale of 1:2.5 million published by China Geological Survey in 2001.
Currently, national geological maps with a scale of 1:500,000 are available, while some
local maps have the scale to 1:50,000. We need to fully consider such resources to improve
the permeability/hydrolithology products in China in terms of both horizontal resolution
and available depth. This is critical for building a more reliable hydrostratigraphy, which
could substantially improve model performance. In addition to permeability, any regions
with more detailed local measurements should also be utilized to evaluate and refine
current modeling formulations.
(5) Building large-scale hydrologic models using different formulations is encouraged. Model
comparisons are necessary to identify the strengths and limitations of different modeling
systems on focus issues (Bailey et al., 2016; Zafarmomen et al., 2024; Kim et al., 2008).
Such community activities are also helpful in reaching consensus on critical questions,
such as conceptual models or model parameterizations, calibrations, evaluations, and
opportunities incorporating new techniques and concepts. All of these factors are essential
for improving the performance of next-generation models in China and can provide
valuable insights for modeling efforts in other parts of the world.

## Data and code availability

Datasets we used are all from public sources and have been cited in the main text. The ParFlow
Version 3.12–3.13 we used can be found here: https://doi.org/10.5281/zenodo.4816884

## Author contributions

Conceptualization: CY and RM. Methodology: CY, LC, and RM. Investigation: CY, ZJ, WX,
and RM. Resources: ZW, XZ, YZ, YD, and RM. Writing – original draft: CY. Writing – review
and editing: CY, JM, and RM.

## Competing interests

The authors have declared that there are no competing interests.

## Acknowledgements

We are pleased to acknowledge that the work reported in this paper was substantially performed using the Princeton Research Computing resources at Princeton University which is a consortium of groups led by the Princeton Institute for Computational Science and Engineering (PICSciE) and the Office of Information Technology's Research Computing. We would like to acknowledge the high-performance computing support from Derecho provided by NCAR's Computational and Information Systems Laboratory.

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

A High-Resolution Global Hydrography Map Based on Latest Topography Dataset, Water
Resour Res, 55, 5053-5073, 10.1029/2019wr024873, 2019.
Yan, F., Shangguan, W., Zhang, J., and Hu, B.: Depth-to-bedrock map of China at a spatial
resolution of 100 meters, Sci Data, 7, 2, 10.1038/s41597-019-0345-6, 2020.
Yang, C., Maxwell, R., McDonnell, J., Yang, X., and Tijerina-Kreuzer, D.: The Role of Topography
in Controlling Evapotranspiration Age, Journal of Geophysical Research: Atmospheres, 128,
e2023JD039228, https://doi.org/10.1029/2023JD039228, 2023a.
Yang, C., Tijerina-Kreuzer, D. T., Tran, H. V., Condon, L. E., and Maxwell, R. M.: A high-
resolution, 3D groundwater-surface water simulation of the contiguous US: Advances in the
integrated ParFlow CONUS 2.0 modeling platform, J Hydrol, 130294,
https://doi.org/10.1016/j.jhydrol.2023.130294, 2023b.
Yang, C., Li, H.-Y., Fang, Y., Cui, C., Wang, T., Zheng, C., Leung, L. R., Maxwell, R. M., Zhang,
Y.-K., and Yang, X.: Effects of Groundwater Pumping on Ground Surface Temperature: A
Regional Modeling Study in the North China Plain, Journal of Geophysical Research:
Atmospheres, 125, e2019JD031764, 10.1029/2019jd031764, 2020.
Yang, J. and Huang, X.: The 30 m annual land cover dataset and its dynamics in China
from 1990 to 2019, Earth Syst. Sci. Data, 13, 3907-3925, 10.5194/essd-13-3907-2021, 2021.
Yang, Y., Feng, D., Beck, H. E., Hu, W., Sengupta, A., Monache, L. D., Hartman, R., Lin, P., Shen,
C., and Pan, M.: Global Daily Discharge Estimation Based on Grid-Scale Long Short-Term
Memory (LSTM) Model and River Routing, ESS Open Archive, DOI:
10.22541/essoar.169724927.73813721/v1, 2023c.
Yin, Z., Lin, P., Riggs, R., Allen, G. H., Lei, X., Zheng, Z., and Cai, S.: A synthesis of Global
Streamflow Characteristics, Hydrometeorology, and Catchment Attributes (GSHA) for large
sample river-centric studies, Earth Syst. Sci. Data, 16, 1559-1587, 10.5194/essd-16-1559-2024,
2024.
Yu, X., Luo, L., Hu, P., Tu, X., Chen, X., and Wei, J.: Impacts of sea-level rise on groundwater
inundation and river floods under changing climate, J Hydrol, 614, 128554,
https://doi.org/10.1016/j.jhydrol.2022.128554, 2022.
Zafarmomen, N., Alizadeh, H., Bayat, M., Ehtiat, M., and Moradkhani, H.: Assimilation of
Sentinel-Based Leaf Area Index for Modeling Surface-Ground Water Interactions in Irrigation
Districts, Water Resour Res, 60, e2023WR036080, https://doi.org/10.1029/2023WR036080,
2024.

Zhang, J., Condon, L. E., Tran, H., and Maxwell, R. M.: A national topographic dataset for
hydrological modeling over the contiguous United States, Earth Syst. Sci. Data, 13, 3263-3279,
10.5194/essd-13-3263-2021, 2021.

Zhang, Y., Kong, D., Gan, R., Chiew, F. H. S., McVicar, T. R., Zhang, Q., and Yang, Y.: Coupled
estimation of 500 m and 8-day resolution global evapotranspiration and gross primary
production in 2002–2017, Remote Sens Environ, 222, 165-182,
https://doi.org/10.1016/j.rse.2018.12.031, 2019.

Zhao, K., Fang, Z., Li, J., and He, C.: Spatial-temporal variations of groundwater storage in China:
A multiscale analysis based on GRACE data, Resources, Conservation and Recycling, 197,
107088, https://doi.org/10.1016/j.resconrec.2023.107088, 2023.
