# Peer review of "CONCN: A high-resolution, integrated surface water-groundwater"

_Hydrology and Earth System Sciences, 2024_

## Community Comment (CC2)

1) How did the authors handle uncertainty in datasets for potential recharge and soil properties in regions with sparse observational data, particularly in arid and semi-arid zones? Could more details on uncertainty quantification be provided?

Uncertainties of the large-scale hydrologic modeling are largely determined by the uncertainties in the data products used. The generation of the input datasets is always a huge amount of work and separated from the modeling, i.e., the dataset generation and the large-scale hydrologic modeling are the focuses of two different communities and this division will be clearer moving forward in the big-data era. We mentioned that if it is five years earlier, such a modeling is impossible as many global data products were not available. As one of the most important efforts in our modeling, we tried to choose the best available datasets at current stage to reduce potential uncertainties. However, quantifying the intrinsic uncertainties in these data products are out of the scope of our work. Future work incorporating local available data is necessary to further improve the quality of the input datasets or decrease the uncertainties in them. One of our goals in this regard is to keep an eye on the advances in relevant data products in the community and dynamically replace some of the inputs with those of higher qualities at a feasible frequency.

Regarding the selection of datasets in our modeling, we have lengthy discussions for both potential recharge and soil properties. Please refer to lines 204 to 233 and lines 235 to 270, respectively. We also briefly summarized them here as below.

As we mentioned in the manuscript, our objective is to continuously improve the workflow of large-scale surface water-groundwater modeling using ParFlow for community use globally. Therefore, we started from the workflow of CONUS 2.0. Then we found replacements of some datasets, e.g., those existing in US but are absent in China, or those having better ones in China area. For soil texture and deep geology, we used the same datasets GSDE and GLHYMPS 1.0. For flow barriers, there is a better data product for China area, so we replaced the global one by the new one. Also, all of them are the datasets well recognized by the community, i.e., the best choice we can use in China area not only because of CONUS 2.0 using them. Additionally, the combination of these datasets showed outstanding performance when they were tested in three large basins (the Upper Colorado River basin, the Little Washita basin, and the Delaware-Susquehanna Basin) based on ParFlow simulations evaluated by observed streamflow and water table depth.

For potential recharge (P-ET), we compared those generated by different precipitation and ET products and further constrained them with prior knowledge. We collected the latest P and ET products with higher spatial resolutions and long enough durations. Then we further filtered out those contrasting to prior knowledge. This is easy to do as

it is well-known that P and ET products are of high uncertainties. For example, we know there is recharge in the upstream of Heihe River Basin, so the combinations of P and ET generating zero or negative potential recharge in this area will not be considered any more. In manuscript, we also highlighted such significant uncertainties in the products challenging both the data and modeling communities. We also provided a possible solution in future work to generate P and ET products under a unified modeling framework constrained by the water balance.

2) The CONCN 1.0 model covers a vast area at high resolution, which demands substantial computational resources. Could the authors discuss any measures taken to optimize computational efficiency and how the model's scalability could be extended to similar hydrologic regions?

Yes. We used seepage face as the top boundary condition in the first phase of the spinup and then turned on the overland flow in the second phase. This avoids the meaningless surface water-groundwater exchange in the early stage which mainly stabilizes the groundwater. For the scalability, we also have some experience. The CONCN model and the CONUS 2.0 model have very similar dimensions. Therefore, they take approximately the same wall clock time for spinup. Yet due to the larger area of arid and semi-arid regions in China, where the on and off of overland flow (integrated or groundwater only) may take more time to converge. Thus, the spinup of CONCN model takes slightly longer time. Additionally, ParFlow has excellent parallel scalability for different domain sizes and heterogeneities, which has been carefully tested and discussed in Ashby and Falgout (1996).

3) Would the authors consider using coarser resolution or data assimilation techniques to make the model more computationally accessible, particularly for policy-making applications?

Might be a choice but it is really hard to say that this is what we expect. Coarse resolutions will miss a lot subgrid variations which will cause the deviations of the simulation results. This is a well-known issue in the community of earth system modeling. A model with a higher resolution generally shows better performance if the parameterization is reliable enough (or similar). Thus, we are trying to build a high-resolution model to ensure the model performance instead of moving backward.

4) I recommend that the authors consider including a comparison with data assimilation approaches to enhance model accuracy and reduce uncertainties, especially in data-scarce regions. Data assimilation has been effectively applied in hydrologic modeling to integrate observed data with model predictions, often improving the alignment with real-world conditions. Techniques like Kalman filters or

variational data assimilation could complement the current workflow, particularly for improving estimates of potential recharge and water table depth in arid and semi-arid regions where observational data is limited. A comparison with data assimilation methods may also highlight the strengths of the CONCN model and provide a pathway for future enhancements in large-scale hydrologic modeling.

Data assimilation is an efficient approach for incorporating observations and doing parameter inversion. This is in our future plan of our modeling platform. The foundation of a sustainable modeling platform is to build a model of reliable/acceptable performances, i.e., the very first thing. Then the strength of the data assimilation can be fully leveraged. As a result, our first step focuses on the model structure, data selection, spinup, evaluation etc. We also identified the challenges in the modeling. It has been a huge step from scratch, and costs more than two years involving all authors and other collaborators. We are not to build a perfect modeling platform with everything in one paper which is impossible in a short time. We are doing step by step to gradually improve the modeling platform and timely share the results of each step with the community.

Once there is a model with acceptable performance, not only data assimilation but also many other approaches, e.g., emulators, could be incorporated into this modeling framework. The data assimilation improves some of the parameters relying on the observations of some others. This means it still has high requirements of observations. As mentioned by reviewer, it is data scarce in arid and semi-arid areas. Collecting long-term observations of enough spatial density, e.g., water table depth, which has been confirmed useful in data assimilation of groundwater modeling, is a big challenge. In some regions, it is even impossible as the observation network has not been built. Therefore, we also discussed in the manuscript that, moving forward, this modeling platform needs collaborative efforts from different communities.

5) To strengthen the contextual foundation of this study, I recommend the authors cite established integrated hydrologic models like SWAT-MODFLOW in the introduction. SWAT-MODFLOW, widely used for its integration of surface and subsurface processes, has significantly advanced our understanding of coupled surface-groundwater systems across various scales. Citing SWAT-MODFLOW alongside ParFlow and other large-scale models would provide readers with a broader perspective on the tools available for integrated hydrologic modeling. This comparison may also underscore the unique challenges and innovations of applying ParFlow within China's hydrologic and geologic context, while highlighting the importance of diverse model approaches for managing complex water resources.

I strongly recommend to cite below paper:

"Assessing regional-scale spatio-temporal patterns of groundwater–surface water interactions using a coupled SWAT-MODFLOW"

"Assimilation of sentinel-based leaf area index for modeling surface-ground water interactions in irrigation districts"

"Development and application of the integrated SWAT–MODFLOW model."

Our objective is in the framework of large-scale hydrologic modeling. We have done substantial literature review and listed the latest large-scale hydrologic models either in China or at global/national scale, including those using MODFLOW. Though SWAT-MODFLOW is relevant to integrated hydrologic modeling, these three papers are neither relevant to China nor to global/national scale. We fully respect reviewer's strong desire to cite the new published WRR paper, so we cite all three papers in the discussion. Please refer to line 508 in the revised manuscript.

Overall, we appreciate reviewer's interesting thoughts and are pleasant to exchange our ideas on these thoughts. However, these thoughts are more or less deviated from the objective of this work or beyond the scope of this very first and important step. Open questions remain in the large-scale high resolution groundwater modeling. As we mentioned in the discussion, all three groundwater models show different water table depths implying large uncertainties. This is more challenging in an integrated framework and in a data-poor region. Therefore, we are trying to use what we can use in such a region to build a model with acceptable performances (actually, it is unexpected excellent performances) as a reference for the community. The data/datasets collection, selection, processing, assembling the model, fetching computational resources, running the model, and analyzing the simulation results and comparing them with previous models, etc., have been substantial work. We don't aim to finish everything in one step, but to gradually improve it with time.

---

## Author Comment (AC1)

**General comments**

The authors present a new hydrologic modeling platform over continental China, based on the ParFlow model, aimed at providing information for surface water and groundwater resources management. Their setup is adapted from the CONUS 2.0 modeling platform over the US. The authors discuss the parameters and input data used and they provide a comparison with modeled and observed data for groundwater table depth and river discharge.

The paper is well written and pleasant to read. The results are interesting and well presented via clear figures. My main concern is about the comparison with other datasets, which may be a little too simple, as detailed below.

We are tankful to the reviewer for all the constructive comments which substantially improve our manuscript. We have addressed them point by point below and revised the manuscript at the corresponding locations.

In particular:

- l. 278-293: Do I understand correctly that for the spinup, the authors used the 1981-2010 P-ET average as constant atmospheric forcing until a quasi-steady state was reached? Is this resulting state used for the evaluation in the next section or have the authors simulated a transient run over 1981-2010, starting from this quasi-steady state? This is not clearly stated, while it is very important for the evaluation and its analysis. For the following comments, I will assume that the authors evaluate the resulting quasi-steady state against other data sets.

  Yes, it is a quasi-steady state model forced by the average P-ET of 1981-2010. We clarified it, please refer to lines 293-294 in the revised manuscript.

- section 4.1. and 4.2.: While the main motivations for this modeling platform are (1) the impacts on water resources of the increased frequency, intensity, and duration of extreme weather events and (2) the management of these water resources, e.g., to prevent water scarcity, the authors limit their evaluation to a comparison of the steady state, which represents an idealized situation that never happens in the real world. In particular, the ability of the modeling platform to represent the dynamics (temporal evolution) on a yearly or better monthly or even daily time scale is not considered in this study, while this would be essential to assess whether the modeling platform is able to meet its primary aim (i.e., the aforementioned motivations).

Reviewer well summarized some of the main motivations to develop this modeling platform. This work is the very first and very important step of this modeling platform. The current model is not the whole thing. This first step aims to build the foundation of the modeling platform, focusing on the model structure, parameterizations, data selection and processing, model assembling and spinup, observation data collection and cleaning, comparison of model formulation and simulation results with other models or datasets, and identify the challenges and requirements to move forward (beyond the motivations summarized above). This step costs a team of more than 10 people more than 2 years (all authors and others not listed).

A steady state model representing a long-term average state is important to demonstrate the general reliability of the current modeling formulation in the target area and unravel the remaining deficiencies in the modeling community. This work is an important reference and/or inspiration for the large-scale hydrologic modeling community. This steady state model will be the starting point for the transient run. Starting from a multi-year averaged state will generally reduce the spinup time of the transient run for a specific year. It is computational expensive to run the model, so we first build the steady state model and then run selected years when needed, according to the requirement of focused objectives. One more reason is we will run the transient state model using ParFlow coupled with the latest Common Land Model. The latest Common Land Model has increased functionalities and could be helpful to better understand the hydrologic cycle in China. The workflow and the necessary data based on the new coupling model are under preparation, which is again a huge amount of work.

- section 4.1.
    - esp. l. 347-348: Do the authors use the longest available period for each gauge or the longest overlapping period (i.e., max 9 years between 2002 and 2010)? In any case, this relies on the hypothesis that an observed average over a few years (sometimes even only two years) as well as an observed average over two to several years covering another period (2002-2021) is representative for a steady state based on 1981-2010. I am not convinced that this hypothesis is true. I could agree that, the longer the observation period is, the closer the average gets to a steady state over the same period, even if this should still be verified. But in my opinion, there is no guarantee that the average over 2002-2021 is representative for the 1981-2010 steady state as this ignores potential shifts in the terrestrial water regime, e.g., due to climate change. One could think of the impact on

streamflow of earlier snow melt, less snow accumulation in winter, more frequent extreme events, multi-year droughts, etc. Assuming that an average over a few years is representative for a steady state might be even more questionable, since these few years could be characterized by extreme events (droughts, floods). The resulting average would certainly not correspond to a steady-state, preventing a robust comparison. The authors already mention the potential impact of hydraulic engineering (e.g., dam operations) on the average streamflow (see l. 354), especially over short time periods.

We used all data available during 2002-2021. We fully understand the reviewer's concern. This is exactly one of the biggest challenges we encountered in the modeling in China. Collecting, cleaning, and processing observations are the most time-consuming part in our modeling. In China, we don't have a fully-open, public access to observations of streamflow and water table depth. We contacted many people or institutions and gained little. Also, the monitoring networks in China started very late, e.g., the groundwater monitoring network started with a small number of wells (~900) from 2005. Although the streamflow is slightly earlier, it is not as earlier as that of USGS which could date back to 1900s. The only way we can get observations is to digitize them from the yearbook which again is very time consuming. The final scheme used in our paper are the best we can do at current stage considering the normal/acceptable duration for academic outcomes. We highlighted in the manuscript that this challenge largely hampers our modeling and expect the conversations and collaborations with the data monitoring community. This is also one of motivations of this modeling work. Data sharing or public access is urgent to break this bottleneck and may need policy support. More importantly, this is a modeling platform of dynamic efforts, and the current work is the first step. We have collected more data after the submission and will clean and process the data and incorporate them into the evaluation in future work. Please refer to lines 331-338, 390-400, 486-505, and 516-532 in the revised manuscript for the relevant discussion here.

- section 4.2.
  - the structure of this section might be improved to make it easier for the reader to follow. For example, after the first paragraph (l. 364-371) describing Figure 6, one would expect a first analysis of these results. But this analysis only starts on l. 432.

Revised. We plot Figures 6c and 6d in a new Figure 9.

We understand the reviewer's concern. The original logic here is that we first introduce all the materials we used to evaluate the model, i.e., the global datasets, the observations, and the GRACE data. Then we first analyzed the scatterplots of simulations vs. observations, then the residuals vs. GRACE data, and finally the uncertainties remaining in the groundwater models in the community (i.e., Figures 6c and 6d). We may plot Figures 6c and 6d in a new figure after figure 8, but it may prevent audience to compare 6c and 6d with 6a and 6b. We also tried to analyze some after Figure 6, but it is hard to get general conclusions before the analysis of scatterplots and GRACE data.

o   l. 364 and Figure 6: Are the steady states of the two global datasets over the same period as for CONCN (i.e., 1981-2010)? If not, is the hypothesis valid that these steady states, which might have been reached under different climatic conditions, are comparable? For example, if one region experiences less (or more) precipitation and/or higher evapotranspiration due to climate change, the resulting steady state will very likely be different.

Clarified. Please refer to lines 382 and 385.

Additional thoughts please refer to the response to the comment '*l. 259: Why did the authors use the period 1981-2010 and not, e.g., 1991-2020? This could have made the evaluation easier, as the authors state further below that more observations are available for the last years (esp. since the 2000s).*' in this letter.

o   l. 397-416: In the same way as my comments above for the evaluation of streamflow, I do not see any reason why one could assume that the observed average over 2018 could be considered as representative or close to a steady state generated with data from 1981-2010. Especially for water table depth with a potentially huge impact of inherited conditions from previous years (memory effect), not only 2018, but also the previous years would need to be close to the 1981-2010 average hydrologic regime to – maybe – approach a steady-state-like state. I understand that this is the reason why the authors try to strengthen their evaluation with the analysis of the residuals in the context of the long-term trend from GRACE, thereby

trying to make a link between the steady state based on 1981-2010 and 2018, but, in my opinion, the uncertainty of this whole evaluation remains high. The only way to provide a robust and representative evaluation is to do the comparison over a common period.

Please refer to the response to the evaluation of streamflow above for the first part of this comment. Additionally, we used 2018 as it is the first year of the expanded national groundwater monitoring network (> 8000 wells). A much fewer wells (~900) are included in the earlier monitoring network and are mainly distributed in the east China. It is hard to balance the quantity and duration of the observations. Considering the slow variations of groundwater (i.e., the long correlation/memory), we finally used 2018 of more wells. The comparison of residuals with GRACE is an additional approach to evaluate the model and highlights the uncertainties in existing groundwater models in the community. Yes, again, we recognize the mismatch of the durations between simulations and observations is a concern, yet this is the best we can do at current stage.

o l. 401-404: This might be easier to understand if the authors could briefly explain why this analysis integrating GRACE data is needed.

We aim to evaluate the model use multi-source of data generated by different approaches, especially in such a data poor region. Multi-source data can provide cross-evaluation to improve the reliability of the modeling.

o l. 411: if the global models are calibrated, do they not implicitly account for human interaction, via the observational data used for calibration? This would then be contradictory with the statement in l. 406.

The global models we cited were calibrated based on observations without explicitly considering human activities, e.g., groundwater pumping. Their calibrations were done based on observations mainly in America and Europe instead of China. This is also due to the data scarcity in China. Then the calibrated models generated the simulation results we used in our study which were not constrained by the observations in China. This is the first time to evaluate their results with observations and GRACE in China area.

o l. 421-422: In l. 377, I understand that these grid cells are excluded from this analysis (precisely for the reason explained here). Please clarify this.

Clarified. Please refer to line 391 in the revised manuscript.

**Specific comments**

- l. 54-55: While I agree that it is pressing to develop a modeling platform accounting for it, this statement suggests that CONCN accounts for water quality control, which is not the case.

  We also have the particle tracking system which can simulate water ages and have implications for water quality. This is also a component that will be added into the modeling platform. To avoid confusing the audience at current stage, we removed the water quality in the revision.

- l. 99: About the "unique dramatic topographic relief". On one side, each part of the world has a unique relief, thus I could agree with this formulation. On the other side, many other regions (e.g., the US, South America, Africa, Europe, New Zealand, Japan, etc.) have transitions from mountains to coastal plains, thus facing similar challenges for hydrologic modeling.

  Revised.

- Figure 1: What is the meaning of the white coloring within the model domain in Fig. 1f? Here, I would interpret it as "no data", is that correct? If it is zero, it should be colored according to the color bar (i.e., dark blue). If it is "no data", how do the authors deal with it as source-sink term for ParFlow? This should be clarified in the text.

  Clarified. These areas have P-ET of 0 in the model. We cannot show them in log plot.

- l. 167-169: The procedure is not clear to me. Did the authors generate D8 connectivity slopes in addition to the aforementioned D4 slopes? If yes, why was it needed? What do they mean by "vector networks"?

  We generated D8 networks as the input of priorityflow to generate the final D4 networks we need. We compared the D8 networks we generated with the

vector network generated from the higher resolution MERIT Hydro to avoid obvious errors in the inputs of priorityflow.

- l. 180: How are the sinks handled? Is the inflowing ponding water removed before/after each time step?

Yes. A specific key in ParFlow did this automatically.

- l. 200: Why do the authors derive the soil texture from this global dataset instead of using directly the soil hydraulic properties?

Could be an option. Two more reasons: 1) wanted to keep consistent with the CONUS 2.0 workflow and 2) our further processing will generate soil textures fewer than the original data which can help the convergence of the model. This has been tested in our North China Plain model in previous studies.

- l. 208: What is meant by "flow barriers"? Is the permeability in these grid cells further reduced by a factor or set to a very small value?

Yes. Clarified. Please refer to line 218.

- l. 250: "several locations" It might be useful to add some more details here: How many locations? Are they distributed over the country and/or hydroclimatic regions to ensure a representative analysis and selection of datasets?

This is a qualitative filtering. We mainly focused on the following two locations together with other judgements without a specific location. If the P-ET is negative in the top rectangle or the precipitation is not obviously higher than areas around in the bottom rectangle, we filtered out that combination of P-ET. This is easy to do as it is well known that the P and ET data products, especially ET, are of high uncertainties. This is a common challenge in the community.

[Figure]

(f) Potential recharge (mm)

- l. 259: Why did the authors use the period 1981-2010 and not, e.g., 1991-2020? This could have made the evaluation easier, as the authors state further below that more observations are available for the last years (esp. since the 2000s).

We tried to develop a steady-state model that can represent the steady state. This requires the long enough P and ET products in a period without intensified human activities (i.e., better before 1950). This is easy to realize in data-rich US but a challenge in China. If we develop a model forced by P-ET of 1991-2020, the P and ET have been affected by human activities. However, the human activities and the uncertainties are even harder to quantify and represented in the model, which will make the evaluation of the model harder.

- l. 275: It might be useful to add a source or some additional explanation on how Manning's coefficient is "set to vary by land cover type".

Clarified. Please see lines 285-287.

- l. 284: What is the role of the seepage face boundary condition?

This is to speed up the spinup. In the early stage of the spinup, the state of groundwater is far from the final quasi-steady state, so the interactions between groundwater and surface water are meaningless. Hence, we used seepage face instead of overland flow at the beginning to reduce computational load. When the groundwater is almost steady (the river channels are generated), we turn on the overland flow. Please refer to lines 301-302.

- l. 284-285 and 287: On which time scale does the total storage change have to be less than 1% (resp. 3%)? Is it e.g., between two consecutive time steps or on an inter-annual basis?

  Either is fine. I did the latter one.

- l. 292: I understand that it is important to reach an equilibrium for groundwater and for river discharge, but does a quasi-steady state for discharge in arid and semi-arid regions really make sense? Is the resulting discharge not too far away from reality? I would guess that in reality, the discharge is highly variable in these regions, with very low flow, or even no flow at all, most of the time alternating with high discharge after precipitation events or snow melt.

  Good point. Actually, the current modeling has bigger challenges than reviewer's concern as the large intrinsic uncertainties in P and ET datasets, especially in ET. The simulation results are unsatisfied in these areas such as the Endorheic and the Hai River Basins, which we highlighted in the discussion of Figure 5.

- Figure 3: How can the authors explain that they have streamflow values everywhere and not just in the streambeds in the south-east and north-east of the model domain? Or is it just an impression due to the visualization of a dense hydrographic network?

  Partly, yes. Additionally, the surface water and groundwater shared the same head in the top layer. Therefore, the pressure used to calculate the streamflow is actually 0.05 m below the land surface. Therefore, in areas with water table depths smaller than 0.05 m, there are also 'streamflow'.

- Figure 3: It would be useful to add in the caption which period is shown. Or is it the end of the spinup (i.e., resulting quasi-steady state)?

  Added

- Figure 5: It might be useful to add in the caption that the gauges are grouped per basin as shown on Fig. 1b.

  Added

- Figure 7 and in the text: Do the "residuals" correspond to the difference between CONCN and the observed values at the wells?

Clarified.

- l. 453: All regions in the world experience increasing extreme weather events such as droughts and floods. What may make China "one of the most significant ecohydrologic hotspots in the world" could be the intense water use in the highly populated areas of the country. However, this is not accounted for in the model platform presented in this paper.

  Yes, water use is important. This work is the very first step of this modeling platform and we will consider water use in the future work.

**Technical corrections**

- l. 29: Meaning of RSR?

  We clarified it in lines 124-127. As it is too long to explain it in the abstract.

- l. 56: Correct " with a 10 km resolution".

  Corrected.

- l. 67: Meaning of USGS?

  Clarified.

- l. 112: Correct "key components of the ParFlow model"?

  Corrected.

- Figure 1: The north-eastern edge of the domain is hidden behind the color bars.

  Revised.

- Figure 1: In the caption, what do "f.g.", "sil.", and "c.g." stand for?

  Clarified.

- l. 154-156: For clarity, it might be good to specify that this concerns each grid cell individually, e.g., something like "D4 connectivity means that, within each grid cell, streamflow is allowed...".

  Corrected

- l. 182: Correct "with those in IHU"?

  Corrected

- l. 199: In l. 125, the thickness of the second layer (from the top) is 0.3 m. Here, it is indicated to be 0.4 m.

  Corrected

- l. 260: Correct "Tarim River Basin"?

  Corrected

- l. 267: It is important to expand the acronym of CLM to avoid any confusion, as nowadays CLM usually means "Community Land Model", while the CLM integrated in ParFlow is the "Common Land Model".

  Clarified

- l. 340-343: There is a mismatch in the number of gauges: 95 (total) – 6 (no location) – 1 (close to another) – 1 (outside of domain) = 87, not 88.

  Corrected. It should be 95-5-1-1=88.

- Figure 7: Correct "The background shows the average decrease of groundwater storage".

  Corrected

- l. 397: Correct "by the three models"?

  Corrected

- Figure 8: Indicate in the caption that you compare the steady state over 1981-2010 with observations from 2018.

  Indicated

- l. 433: Correct "and the two global models"?

  Corrected

- l. 435: Correct "across the three models"?

Corrected

- l. 445: Correct "below – these require" or maybe "below. These require"?

Corrected

- l. 515: Correct "have been cited"?

Corrected

- l. 524: Correct "reported in this paper"?

Corrected

- l. 525: Correct "which is a consortium"?

Corrected

- l. 526: Correct "and the Office"?

Corrected

---

## Author Comment (AC2)

Yang et al. present results for a coupled surface-groundwater model for continental China. The work is appropriate for HESS, the research is very interesting and of high quality, and the manuscript is well written. Nonetheless, I have several recommendations regarding the evaluation of the model(s). I believe that a more process-oriented evaluation would be more meaningful for both the authors and the readers. My main other concern right now is a lack of discussion of the results. Both aspects are straightforward to rectify though.

Sincere thanks to Professor Wagener's kind words, constructive comments, and insightful thoughts on our work. We carefully read the suggested papers, rethought our work relevant to the concerns, and revised our manuscript to improve it. Here, we simply respond how we address each comment while details of revision can be seen in the final revised manuscript (which cannot be attached with the response).

Larger Comments:

[1] The use of a scaled statistical error metric: The authors state that "Note that all performance evaluations in this paper are based on the RSR value which is the ratio of the root mean squared error to the standard deviation of observations. An RSR value of 1.0 suggests good performance while 0.5 suggests excellent performance (O'neill et al., 2021)." These qualitative statements go back to the paper by Moriasi et al. (2007, doi.org/10.13031/2013.23153) who suggested some subjective qualification for normalized statistical metrics. The use of this subjective language persists even though it has been shown multiple times that the ease with which such values can be achieved varies with system properties (e.g. DOI: 10.1002/hyp.6825; doi.org/10.5194/hess-23-4323-2019). Therefore, these statements of good or poor performance with fixed thresholds are very unhelpful because – depending on the system modelled – it will be easy or hard to achieve these values. Personally (the authors do not have to share this view), I find it much more helpful to assess which system properties allow for high or low model performances (e.g. DOI 10.1088/1748-9326/abfac4 Figure 3 or DOI 10.1088/1748-9326/ad52b0). Such analyses are particularly valuable when done across multiple models, which often show that many models work well under specific conditions (often high wetness levels).

Thanks for this insightful suggestion. We reviewed these mentioned papers and agreed with the concern here. Audience should be cautious to treat these values as absolute performances. The RSR values shown in the manuscript are not comparable between different variables (e.g., drainage area, streamflow, water table depth). They are also not comparable with other case studies evaluating other systems or even the same system but in different periods. Yet, insights of relative performance could be gained from Figure 7 as the same benchmark (observations) is used for evaluating the same behavior (long-term average performance) in generally the same simulation period.

We first added an overall clarification, following the definition of RSR, about the limitation of using RSR as discussed in these listed papers. Then we added the variations of residuals of water table depth with critical factors into the paper, which is, essentially, also the response to comment [2].

[2] Possibility for understanding process controls: The focus on statistical metrics and maps for the comparison of the model with observations or other models provides limited insights into how and (potentially) why the models differ. A simple but effective way to provide more insight is to plot the water table depth (WTD, or other output variables) against (potentially) controlling

variables as functional relationships. For example, when plotting WTD against topographic slope for two of the models used by the authors – GLOBGM and Fan, the recent study by Reinecke et al. (doi.org/10.1088/1748-9326/ad8587) showed that GLOBGM is strongly correlated with slope, while the Fan model and global observations do so much less. Also, the Fan model shows distinct WTD differences between water and energy limited regions, while GLOBGM hardly does so. Similarly to my point 1, what controls the variability of model outputs and the output differences? These plots would include data, which the authors should have readily available – hence there is not much additional effort needed to try this.

Thanks for this constructive comment. We do have a substantial discussion about the shallowed simulated water table depth and the uncertainties caused by human activities. It's unfortunate that they were buried in the original manuscript probably due to the limitation of the manuscript structure as mentioned by Reviewer in comment [5]. In the revision, we added the variations of residuals with key factors (e.g., elevation, slope) into the manuscript (following figure 3 in DOI 10.1088/1748-9326/abfac4 or figure 9 in doi.org/10.5194/gmd-12-2401-2019) and reorganized the paper structure of relevant sections to better deliver our points. Yes, this is also a response to comment [1].

[3] Model omissions: Over 0.5 million km2 of Southern China has Karst geology (doi.org/10.1007/s10980-019-00912-w), which shows significantly different recharge patterns than many other geologies (doi.org/10.1073/pnas.1614941114). How is this reflected in the model set-up? Do these regions show distinctly different patterns than other areas regarding recharge or other variables?

Good point. Previous studies using ParFlow in Karst regions, such as the entire continental US (Yang et al., 2023, doi.org/10.1016/j.jhydrol.2023.130294) and the individual watershed in Florida (Srivastava et al., 2014; doi.org/10.1016/j.jhydrol.2014.10.020) show satisfied performances. Therefore, we didn't take specific actions in such regions. But we fully understand the recharge patterns in Karst regions might be highly different from other regions. The basic idea behind our work is that, at large enough scale, the Karst geology can be assumed as porous media while we recognize that the limitation of this idea must exist. Nevertheless, high hydraulic conductivities were setup in Karst regions in our model. We rechecked the residuals of water table depth shown in Figure 7 in the original manuscript, we do see something special, i.e., deeper simulated water table in all three models in the Karst regions and that GLOBGM v1.0 is the most significant one. We inferred that this might be caused by a larger P-ET in 2018 than long-term average P-ET but we cannot reject that this might be also attributed to the Karst geology. For example, wells are always drilled in places without significant Karst signatures and thus hold normal water table depths. Yet the higher average hydraulic conductivity might cause deeper water table in the simulation. Thanks for this good point motivating us to rethink this important question and we added this additional discussion into the revised manuscript.

[4] Comparison with global models: Global models are rather crude approximations of local hydrology – shown regularly. Comparison to these models is a good starting point, but also limited in what one can learn. Do any national scale modelling efforts exist for China that would also provide a comparison for the model introduced here? Clearly the model presented here has tremendous potential – given its coupled nature – but how would it have to be further improved? It would be interesting to discuss more what additional aspects local or regional models might consider relevant.

We had a lot of efforts regarding this. Unfortunately, we didn't get the results of relevant models. We understand and respect the preferences of authors of these models. As a result, we highlighted in the discussion that model comparison is encouraged. Yet it may take time to build a desired environment of the community.

[5] Lack of discussion: As is often the danger when Results and Discussion sections are not separated, there is a lack of actual discussion. The discussion section should place the results in context of existing literature. This has not yet been done. Other evaluations of the models used exist. Other modelling studies have assessed different strategies for China or globally Etc. The authors need to place their results into such context, preferably by separating Results and Discussion into distinct sections.

We added the new discussion mentioned above into the manuscript and reorganized the structure to make the paper more readable.

Minor Comments:

[6] Line 85ff.: The authors state that "Significant progresses or consensus have been achieved in community discussions regarding model parameterization, evaluation, calibration, and intercomparison". Given that at least the cited Gleeson et al. stresses the current lack of adequate evaluation strategies for global models, I would personally not frame it quite this positively. I do think that there is still significant advancement needed to derive at adequate strategies, and I also think that consensus is not yet there.

Corrected and cited new relevant papers, e.g., Heinicke et al. (ERL, 2024) and Reinecke et al. (ERL, 2024).

[7] Figure 6. The lower plots show positive and negative deviations from 0. The maps would be much clearer if the authors were to use a diverging color scheme as they do in Figure 7.  Though I can also see that the authors prefer to keep the colors similar to the actual values.

Revised.

---

## Author Response (AR1)

**Editor:**

Dear authors,

Your manuscript has received evaluations from two nominated reviewers and a researcher involved in the topic. In addition to comments and requests for clarification on certain aspects of the study, a few things have been pointed out that require some revision. All criticism is constructive and aimed at improving not only the quality of the article but also its readability by a broader readership of HESS. While Reviewer #2 offered a largely positive evaluation, Reviewer #1 expressed some more critical concerns, particularly regarding the scientific quality of the study. This aspect of the article requires the most significant improvement.

Additionally, I would recommend enhancing the presentation of the results to make it more effective, as in certain sections, the flow of the argument appears to be somewhat fragmented, potentially leading to a loss of coherence. In particular, I would like to draw the attention of the senior author of the article (R. Maxwell) to the need for improvement of paragraphs 4.2 and 4.3. Moreover, the conclusion paragraph appears to be a combination of the introduction and a summary of the results rather than real conclusions (although it is not always an easy task to write this final section, I should admit). One potential improvement could be to transfer the concepts expressed in paragraph 4.3, titled "Challenges and opportunities going forward," to the concluding paragraph. Alternatively, and perhaps more effectively, sections 4.3 and 5 could be merged, properly revised, and renamed as "Conclusions and opportunities going forward". The determination of the most appropriate course of action rests with the authors.

The original submission is released under major revisions and, together with the revised version of the article, the authors are required to upload detailed point-by-point replies to the comments received by the reviewers and the community scientist.

Sincere thanks to editor for the nice summary of all the comments and the constructive suggestions. We fully agree that "All criticism is constructive and aimed at improving not only the quality of the article but also its readability by a broader readership of HESS". In the revision, we addressed all the comments from editor, two reviewers, and the community researcher point by point as follows. We are grateful to the reviewers for these constructive comments.

Specifically, we (1) enhanced the presentation of the results, by clarifying the figure titles, fixed the positions of colorbars in Figure 1, added new figures including Figures 9 and 10; (2) reorganized the results and discussion to smooth the flow, by dividing them into the new section 4 for *Simulation and evaluation* and the new section 5 for *Discussion*; and (3) combining the '*Challenges and opportunities going forward*' and the '*Conclusions*' to a new section as '*Summary and going forward*'.

**Reviewer #1:**

**General comments**

The authors present a new hydrologic modeling platform over continental China, based on the ParFlow model, aimed at providing information for surface water and groundwater resources management. Their setup is adapted from the CONUS 2.0 modeling platform over the US. The authors discuss the parameters and input data used and they provide a comparison with modeled and observed data for groundwater table depth and river discharge.

The paper is well written and pleasant to read. The results are interesting and well presented via clear figures. My main concern is about the comparison with other datasets, which may be a little too simple, as detailed below.

We are thankful to reviewer #1 for all the constructive comments which have substantially improved our manuscript. We have addressed them point by point below and made corresponding revisions to the manuscript.

In particular:

- l. 278-293: Do I understand correctly that for the spinup, the authors used the 1981-2010 P-ET average as constant atmospheric forcing until a quasi-steady state was reached? Is this resulting state used for the evaluation in the next section or have the authors simulated a transient run over 1981-2010, starting from this quasi-steady state? This is not clearly stated, while it is very important for the evaluation and its analysis. For the following comments, I will assume that the authors evaluate the resulting quasi-steady state against other data sets.

  Yes, it is a quasi-steady state model forced by the average P-ET over 1981–2010. We clarified it in the revision, please refer to lines 301–303 in the revised manuscript.

- section 4.1. and 4.2.: While the main motivations for this modeling platform are (1) the impacts on water resources of the increased frequency, intensity, and duration of extreme weather events and (2) the management of these water resources, e.g., to prevent water scarcity, the authors limit their evaluation to a comparison of the steady state, which represents an idealized situation that never happens in the real world. In particular, the ability of the modeling platform to represent the dynamics (temporal evolution) on a yearly or better monthly or even daily time scale is not considered in this study, while this would be essential to assess whether the modeling platform is able to meet its primary aim (i.e., the aforementioned motivations).

  The reviewer has effectively summarized some of the key motivations behind the development of this modeling platform. This work represents the first and crucial step in the platform's development. The current model is not intended to be a complete solution but rather serves as the foundation for the platform. It focuses on aspects such as model structure, parameterizations, data selection and processing, model assembly and spin-up, observation data collection and

cleaning, and comparing model formulations and simulation results with other models or datasets. Additionally, it identifies the challenges and requirements necessary for future progress, beyond the motivations outlined by the reviewer. This step has involved more than 10 people and over two years of work (including both listed and unlisted authors).

A steady-state model, representing a long-term average, is critical for demonstrating the general reliability of the current model formulation in the target area and for identifying remaining gaps within the modeling community. This work serves as an important reference and potential inspiration for the large-scale hydrological modeling community. The steady-state model will also serve as the starting point for the transient simulations. Beginning with a multi-year averaged state generally reduces the spin-up time for transient runs of specific years. Given the computational expense of running the model, we have opted to first develop the steady-state model and then run selected years as needed based on the objectives in the future. Furthermore, the transient model will be run using ParFlow coupled with the latest Common Land Model (CoLM). The updated CLM offers enhanced functionality and will help improve our understanding of the hydrological cycle in China. The workflow and necessary data for this new coupling model are still under development, requiring significant additional effort.

- section 4.1.

    - esp. l. 347-348: Do the authors use the longest available period for each gauge or the longest overlapping period (i.e., max 9 years between 2002 and 2010)? In any case, this relies on the hypothesis that an observed average over a few years (sometimes even only two years) as well as an observed average over two to several years covering another period (2002-2021) is representative for a steady state based on 1981-2010. I am not convinced that this hypothesis is true. I could agree that, the longer the observation period is, the

closer the average gets to a steady state over the same period, even if this should still be verified. But in my opinion, there is no guarantee that the average over 2002-2021 is representative for the 1981-2010 steady state as this ignores potential shifts in the terrestrial water regime, e.g., due to climate change. One could think of the impact on streamflow of earlier snow melt, less snow accumulation in winter, more frequent extreme events, multi-year droughts, etc. Assuming that an average over a few years is representative for a steady state might be even more questionable, since these few years could be characterized by extreme events (droughts, floods). The resulting average would certainly not correspond to a steady-state, preventing a robust comparison. The authors already mention the potential impact of hydraulic engineering (e.g., dam operations) on the average streamflow (see l. 354), especially over short time periods.

We used all data available during 2002-2021. We fully understand the reviewer's concern and acknowledge that this is one of the major challenges we faced during the modeling in China. Collecting, cleaning, and processing observational data have been the most time-consuming aspects of our work. Unfortunately, in China, there is limited public access to comprehensive streamflow and groundwater depth data. The monitoring networks have been established relatively recently; for instance, the groundwater monitoring network began in 2005 with only about 900 wells. While streamflow data became available slightly earlier, it still doesn't compare to the extensive, publicly accessible historical data available from organizations like the USGS, which dates back to the 1900s.

The only feasible approach for obtaining a large number of observations has been to digitize data from yearbooks, which, while useful, is a very labor-intensive process. The scheme presented in our paper represents the best possible outcome given the constraints

on time and resources, while also adhering to typical academic timelines. We highlight in the manuscript that this data limitation significantly hampers our modeling efforts, and we emphasize the need for ongoing collaboration with the data monitoring community. This issue is one of the key motivations for our modeling work. We strongly believe that enhanced data sharing and public access are crucial to overcoming this bottleneck, and such initiatives may require policy support.

Importantly, this modeling platform is a dynamic, evolving effort. The current study is the first step, and since the submission, we have gathered additional data, which we will clean and process for future evaluations. Please refer to lines 342–349, 406–416, 534–548, and 575–593 in the revised manuscript for a more detailed discussion of these points.

- section 4.2.
  - the structure of this section might be improved to make it easier for the reader to follow. For example, after the first paragraph (l. 364-371) describing Figure 6, one would expect a first analysis of these results. But this analysis only starts on l. 432.

    Revised. We plotted Figures 6c and 6d in a new Figure 9.

    We understand the reviewer's concern. The original structure follows a logical sequence where we first introduce all the materials used to evaluate the model, including the global datasets, observational data, and GRACE data. Next, we present the analysis in stages: first, scatterplots comparing simulations with observations; second, residuals compared with GRACE data; and finally, the uncertainties that remain in the community's groundwater models (i.e., Figures 6c and 6d).

While we could move Figures 6c and 6d to a new figure after Figure 8, this might make it difficult for the audience to directly compare these figures with 6a and 6b. We also considered analyzing Figures 6c and 6d immediately after Figure 6, but it is challenging to draw meaningful conclusions without first addressing the scatterplots and GRACE data analysis. We finally chose the first option as it looks like a better solution.

o  l. 364 and Figure 6: Are the steady states of the two global datasets over the same period as for CONCN (i.e., 1981-2010)? If not, is the hypothesis valid that these steady states, which might have been reached under different climatic conditions, are comparable? For example, if one region experiences less (or more) precipitation and/or higher evapotranspiration due to climate change, the resulting steady state will very likely be different.

Clarified. Please refer to lines 389–401. The two global datasets have the durations of 2004–2014 and 1958–2015, respectively.

Additional thoughts please refer to the response to the comment '*l. 259: Why did the authors use the period 1981-2010 and not, e.g., 1991-2020? This could have made the evaluation easier, as the authors state further below that more observations are available for the last years (esp. since the 2000s).*' in this letter.

o  l. 397-416: In the same way as my comments above for the evaluation of streamflow, I do not see any reason why one could assume that the observed average over 2018 could be considered as representative or close to a steady state generated with data from 1981-2010. Especially for water table depth with a potentially huge impact of inherited conditions from previous years (memory effect), not only 2018, but also the previous years would need to be close to

the 1981-2010 average hydrologic regime to – maybe – approach a steady-state-like state. I understand that this is the reason why the authors try to strengthen their evaluation with the analysis of the residuals in the context of the long-term trend from GRACE, thereby trying to make a link between the steady state based on 1981-2010 and 2018, but, in my opinion, the uncertainty of this whole evaluation remains high. The only way to provide a robust and representative evaluation is to do the comparison over a common period.

Please refer to the response to the evaluation of streamflow above for the first part of this comment. Additionally, we used the observations in 2018 as it is the first year of the expanded national groundwater monitoring network, which now include over 8000 wells. In contrast, the earlier monitoring network include only about 900 wells, primarily located in eastern China. It is challenging to balance the quantity and duration of the available observations. Given the slow variations of groundwater (i.e., the long correlation/memory), we opted to use data from 2018, which provides a more comprehensive coverage of wells.

It is the same situation in previous studies (Fan et al., 2007 and 2013). Fan compiled the long-term average observations of water table depth or hydraulic head over the period 1927–2005 from 549,616 wells and 81% of them have only one reading.

The comparison of residuals with GRACE data is an additional approach to evaluate the model and highlights the uncertainties in existing groundwater models in the community. As scatterplots only show the overall performance of the model, we further illustrate the spatial distribution of residuals and refine the model's evaluation using the GRACE data.

We recognize the mismatch in the durations of simulations and observations remains a concern. However, this is the best approach we can take at current stage.

- l. 401-404: This might be easier to understand if the authors could briefly explain why this analysis integrating GRACE data is needed.

First, some of the responses above could also be the response here. We copy them here "The comparison of residuals with GRACE data is an additional approach to evaluate the model and highlights the uncertainties in existing groundwater models in the community. As scatterplots only show the overall performance of the model, we further illustrate the spatial distribution of residuals and refine the model's evaluation using the GRACE data."

Additionally, we aim to evaluate the model use multi-source of data generated by different approaches, especially in such a data poor region. Multi-source data can provide cross-evaluation to improve the reliability of the modeling.

- l. 411: if the global models are calibrated, do they not implicitly account for human interaction, via the observational data used for calibration? This would then be contradictory with the statement in l. 406.

The global models we cited were calibrated based on observations without explicitly considering human activities, e.g., groundwater pumping. Additionally, their calibrations were done based on observations mainly from America and Europe rather than China. This is also due to the data scarcity in China. Then the calibrated models generated the simulation results we used in our study which were not constrained by the observations in China. This is the first time to evaluate their results with observations and GRACE data in China

area. We have added discussion on this challenge, namely that two model inputs, hydraulic conductivity (K) and pumping, can have the same impact on water table depths. This is a particular challenge for the community given the uncertainties in subsurface architecture and the lack of extraction data.

- l. 421-422: In l. 377, I understand that these grid cells are excluded from this analysis (precisely for the reason explained here). Please clarify this.

  Clarified. Please refer to line 407 in the revised manuscript.

**Specific comments**

- l. 54-55: While I agree that it is pressing to develop a modeling platform accounting for it, this statement suggests that CONCN accounts for water quality control, which is not the case.

  We also have the particle tracking system which can simulate water ages and have implications for water quality. This is also a component that will be added into the modeling platform. To avoid confusing the audience at current stage, we removed the water quality in the revision.

- l. 99: About the "unique dramatic topographic relief". On one side, each part of the world has a unique relief, thus I could agree with this formulation. On the other side, many other regions (e.g., the US, South America, Africa, Europe, New Zealand, Japan, etc.) have transitions from mountains to coastal plains, thus facing similar challenges for hydrologic modeling.

  Revised.

- Figure 1: What is the meaning of the white coloring within the model domain in Fig. 1f? Here, I would interpret it as "no data", is that correct? If it is zero, it should be colored according to the color bar (i.e., dark blue). If it is "no data", how do the authors deal with it as source-sink term for ParFlow? This should be clarified in the text.

  Clarified in the title in Figure 1. These areas have P-ET of 0 in the model. We cannot show them in log plot.

- l. 167-169: The procedure is not clear to me. Did the authors generate D8 connectivity slopes in addition to the aforementioned D4 slopes? If yes, why was it needed? What do they mean by "vector networks"?

  We generated D8 networks as the input of priorityflow to generate the final D4 networks we need. We compared the D8 networks we generated with the vector network generated from the higher resolution MERIT Hydro to avoid obvious errors in the inputs of priorityflow.

- l. 180: How are the sinks handled? Is the inflowing ponding water removed before/after each time step?

  Yes. A specific key in ParFlow did this automatically.

- l. 200: Why do the authors derive the soil texture from this global dataset instead of using directly the soil hydraulic properties?

  Could be an option. Two more reasons: 1) wanted to keep consistent with the CONUS 2.0 workflow and 2) our further processing will generate soil textures fewer than the original data which can help the convergence of the model. This has been tested in our North China Plain model in previous studies.

- l. 208: What is meant by "flow barriers"? Is the permeability in these grid cells further reduced by a factor or set to a very small value?

Yes. Clarified. Please refer to lines 225–228.

- l. 250: "several locations" It might be useful to add some more details here: How many locations? Are they distributed over the country and/or hydroclimatic regions to ensure a representative analysis and selection of datasets?

  This is a qualitative filtering. We mainly focused on the following two locations together with other judgements without a specific location. If the P-ET is negative in the top rectangle or the precipitation is not obviously higher than areas around in the bottom rectangle, we filtered out that combination of P-ET. This is easy to judge as the P and ET data products, especially ET, are of high uncertainties, which is a common challenge in the community.

[Figure]

- l. 259: Why did the authors use the period 1981-2010 and not, e.g., 1991-2020? This could have made the evaluation easier, as the authors state further below that more observations are available for the last years (esp. since the 2000s).

  We tried to develop a steady-state model that can represent the steady state. This requires the long enough P and ET products in a period without intensified human activities (i.e., better before 1950). This is easy to realize in data-rich US but a challenge in China. If we develop a model forced by P-ET of 1991-2020, the P and ET have been affected by human activities. However, the

human activities and their uncertainties are even harder to quantify and represented in the model, which will make the evaluation of the model harder.

For example, as discussed in the revised manuscript (lines 524 to 528), Fan et al. (2017) simulated the dynamics of water table depth during 2004–2014. They used the ET disturbed by irrigation, which caused the simulated water table depth even deeper than the observations disturbed by groundwater pumping, yet they didn't account for groundwater pumping and irrigation explicitly in the model.

- l. 275: It might be useful to add a source or some additional explanation on how Manning's coefficient is "set to vary by land cover type".

  Clarified. Please see lines 294–296.

- l. 284: What is the role of the seepage face boundary condition?

  This is to speed up the spinup. In the early stage of the spinup, the state of groundwater is far from the final quasi-steady state, so the interactions between groundwater and surface water are meaningless. Hence, we used seepage face instead of overland flow at the beginning to reduce computational load. When the groundwater is almost steady (the river channels are generated), we turn on the overland flow. Please refer to lines 310–311.

- l. 284-285 and 287: On which time scale does the total storage change have to be less than 1% (resp. 3%)? Is it e.g., between two consecutive time steps or on an inter-annual basis?

  Either is fine. I did the latter one.

- l. 292: I understand that it is important to reach an equilibrium for groundwater and for river discharge, but does a quasi-steady state for discharge in arid and semi-arid regions really make sense? Is the resulting discharge not too far

away from reality? I would guess that in reality, the discharge is highly variable in these regions, with very low flow, or even no flow at all, most of the time alternating with high discharge after precipitation events or snow melt.

Good point. Actually, the current modeling has bigger challenges than reviewer's concern as the large intrinsic uncertainties in P and ET datasets, especially in ET. The simulation results are unsatisfied in these areas such as the Endorheic and the Hai River Basins, which we highlighted in the discussion of Figure 5.

- Figure 3: How can the authors explain that they have streamflow values everywhere and not just in the streambeds in the south-east and north-east of the model domain? Or is it just an impression due to the visualization of a dense hydrographic network?

Partly, yes. Additionally, the surface water and groundwater shared the same head in the top layer. Therefore, the pressure used to calculate the streamflow is actually 0.05 m below the land surface. Therefore, in areas with water table depths smaller than 0.05 m, there are also 'streamflow'.

- Figure 3: It would be useful to add in the caption which period is shown. Or is it the end of the spinup (i.e., resulting quasi-steady state)?

Added

- Figure 5: It might be useful to add in the caption that the gauges are grouped per basin as shown on Fig. 1b.

Added

- Figure 7 and in the text: Do the "residuals" correspond to the difference between CONCN and the observed values at the wells?

Clarified.

- l. 453: All regions in the world experience increasing extreme weather events such as droughts and floods. What may make China "one of the most significant ecohydrologic hotspots in the world" could be the intense water use in the highly populated areas of the country. However, this is not accounted for in the model platform presented in this paper.

  Yes, water use is important. This work is the very first step of this modeling platform and we will consider water use in the future work.

**Technical corrections**

- l. 29: Meaning of RSR?

  We clarified it in lines 125–127. As it is too long to explain it in the abstract.

- l. 56: Correct " with a 10 km resolution".

  Corrected.

- l. 67: Meaning of USGS?

  Clarified.

- l. 112: Correct "key components of the ParFlow model"?

  Corrected.

- Figure 1: The north-eastern edge of the domain is hidden behind the color bars.

  Revised.

- Figure 1: In the caption, what do "f.g.", "sil.", and "c.g." stand for?

  Clarified.

- l. 154-156: For clarity, it might be good to specify that this concerns each grid cell individually, e.g., something like "D4 connectivity means that, within each grid cell, streamflow is allowed…".

  Corrected

- l. 182: Correct "with those in IHU"?

  Corrected

- l. 199: In l. 125, the thickness of the second layer (from the top) is 0.3 m. Here, it is indicated to be 0.4 m.

  Corrected

- l. 260: Correct "Tarim River Basin"?

  Corrected

- l. 267: It is important to expand the acronym of CLM to avoid any confusion, as nowadays CLM usually means "Community Land Model", while the CLM integrated in ParFlow is the "Common Land Model".

  Clarified

- l. 340-343: There is a mismatch in the number of gauges: 95 (total) – 6 (no location) – 1 (close to another) – 1 (outside of domain) = 87, not 88.

  Corrected. It should be 95-5-1-1=88.

- Figure 7: Correct "The background shows the average decrease of groundwater storage".

  Corrected

- l. 397: Correct "by the three models"?

Corrected

- Figure 8: Indicate in the caption that you compare the steady state over 1981-2010 with observations from 2018.

Indicated

- l. 433: Correct "and the two global models"?

Corrected

- l. 435: Correct "across the three models"?

Corrected

- l. 445: Correct "below – these require" or maybe "below. These require"?

Corrected

- l. 515: Correct "have been cited"?

Corrected

- l. 524: Correct "reported in this paper"?

Corrected

- l. 525: Correct "which is a consortium"?

Corrected

- l. 526: Correct "and the Office"?

Corrected

**Reviewer #2:**

Yang et al. present results for a coupled surface-groundwater model for continental China. The work is appropriate for HESS, the research is very interesting and of high quality, and the manuscript is well written. Nonetheless, I have several recommendations regarding the evaluation of the model(s). I believe that a more process-oriented evaluation would be more meaningful for both the authors and the readers. My main other concern right now is a lack of discussion of the results. Both aspects are straightforward to rectify though.

Sincere thanks to Professor Wagener's kind words, constructive comments, and insightful thoughts on our work. We carefully read the suggested papers, rethought our work relevant to the concerns, and revised our manuscript to improve it.

Larger Comments:

[1] The use of a scaled statistical error metric: The authors state that "Note that all performance evaluations in this paper are based on the RSR value which is the ratio of the root mean squared error to the standard deviation of observations. An RSR value of 1.0 suggests good performance while 0.5 suggests excellent performance (O'neill et al., 2021)." These qualitative statements go back to the paper by Moriasi et al. (2007, doi.org/10.13031/2013.23153) who suggested some subjective qualification for normalized statistical metrics. The use of this subjective language persists even though it has been shown multiple times that the ease with which such values can be achieved varies with system properties (e.g. DOI: 10.1002/hyp.6825; doi.org/10.5194/hess-23-4323-2019). Therefore, these statements of good or poor performance with fixed thresholds are very unhelpful because – depending on the system modelled – it will be easy or hard to achieve these values. Personally (the authors do not have to share this view), I find it much more helpful to assess which system properties allow for high or low model performances (e.g. DOI 10.1088/1748-9326/abfac4 Figure 3 or DOI 10.1088/1748-9326/ad52b0). Such analyses are particularly valuable when done across

multiple models, which often show that many models work well under specific conditions (often high wetness levels).

Thank you for your insightful suggestion. After reviewing the cited papers, we fully acknowledge the reviewer's concern. We agree that the audience should be cautious when interpreting these values as absolute performance measures. The RSR values presented in the manuscript are not directly comparable across different variables (e.g., drainage area, streamflow, water table depth) nor with other case studies evaluating different systems or even the same system over different time periods. However, Figure 8 offers some insights into the relative performance, as it uses the same benchmark (observations) to access the same behavior (long-term average performance) within a generally consistent simulation period.

To address the reviewer's concern, we first added a clarification regarding the limitations of using RSR, following its definition, as discussed in papers listed by reviewer (see lines 128 to 135 in the revised manuscript). Additionally, we included Figure 10, along with a corresponding discussion on the variations of water table depth with slope (lines 496 to 532 in the revised manuscript), which also serves as a response to the reviewer's comment [2].

[2] Possibility for understanding process controls: The focus on statistical metrics and maps for the comparison of the model with observations or other models provides limited insights into how and (potentially) why the models differ. A simple but effective way to provide more insight is to plot the water table depth (WTD, or other output variables) against (potentially) controlling variables as functional relationships. For example, when plotting WTD against topographic slope for two of the models used by the authors – GLOBGM and Fan, the recent study by Reinecke et al. (doi.org/10.1088/1748-9326/ad8587) showed that GLOBGM is strongly correlated with slope, while the Fan model and global observations do so much less. Also, the Fan model shows distinct WTD differences between water and energy limited regions, while GLOBGM hardly does so. Similarly to my point 1, what controls the variability of model outputs and the output differences? These plots would include data, which the authors

should have readily available – hence there is not much additional effort needed to try this.

Thank you for this constructive comment. We acknowledge the limitation of focusing solely on metrics and maps. We have included some discussion regarding the shallow simulated water table depth and the uncertainties introduced by human activities. It's unfortunate that they were buried in the original manuscript likely due to the limitation of the manuscript structure as pointed out in comment [5].

In the revision, we added the variations of water table depth with slope (Figure 10 along with the corresponding discussion) into the manuscript (following figure 3 in DOI 10.1088/1748-9326/abfac4 or figure 9 in doi.org/10.5194/gmd-12-2401-2019).

We also reorganized section 4 to present our points more clearly. The results and discussion are now separated into two distinct sections "*Simulation and evaluation*" and "*Discussion*". The first section presents the simulation results, the data used for model evaluation, and the comparison with these data. The second section specially addresses the differences among models and explores the mechanisms behind these differences, aiming to improve our understanding of the advances, limitations, and challenges in large-scale groundwater modeling.

This revision is also a partial response to comments [1] and [5].

[3] Model omissions: Over 0.5 million km2 of Southern China has Karst geology (doi.org/10.1007/s10980-019-00912-w), which shows significantly different recharge patterns than many other geologies (doi.org/10.1073/pnas.1614941114). How is this reflected in the model set-up? Do these regions show distinctly different patterns than other areas regarding recharge or other variables?

Thank you for raising this important point. Previous studies using ParFlow in Karst regions, such as those covering the entire continental US (Yang et al., 2023, doi.org/10.1016/j.jhydrol.2023.130294) and an individual watershed in Florida (Srivastava et al., 2014; doi.org/10.1016/j.jhydrol.2014.10.020) have reported

satisfactory performances. As a result, we didn't take specific actions in such regions. However, we fully understand the recharge patterns in Karst regions may differ significantly from those in other regions. The underlying assumption in our work is that, at large enough scales (e.g., grid cells with a resolution of approximately 1 km), the Karst geology can be assumed as porous media. We acknowledge that this assumption has limitations. Nevertheless, we set higher hydraulic conductivities in Karst regions within our model. We rechecked the residuals of water table depth shown in Figure 7 in the original manuscript, we observe something notable: all three models produced deeper simulated water table in Karst regions with GLOBGM v1.0 showing the most significant discrepancy. Initially, we attributed this to a larger P-ET value in 2018 compared to the long-term average. However, we now hypothesize that this might also be influenced by the Karst geology. Specifically, wells are typically drilled in areas without significant Karst features, and thus hold typical local water table depths. However, the higher average hydraulic conductivity in a gird cell of ~1 km resolution may lead to deeper simulated water table in these areas.

We appreciate this valuable feedback, which has prompted us to reconsider this important issue. We have added this additional discussion in the revised manuscript (see lines 454 to 468).

[4] Comparison with global models: Global models are rather crude approximations of local hydrology – shown regularly. Comparison to these models is a good starting point, but also limited in what one can learn. Do any national scale modelling efforts exist for China that would also provide a comparison for the model introduced here? Clearly the model presented here has tremendous potential – given its coupled nature – but how would it have to be further improved? It would be interesting to discuss more what additional aspects local or regional models might consider relevant.

We put considerable effort into this aspect, but unfortunately, we were unable to obtain the results from the relevant models. We understand and respect the preferences of the authors of these models. As a result, we emphasized in the final section (or Section 4.3

in the original manuscript) that model comparison is encouraged, although we recognize that building an ideal collaborative environment within the community may take time.

[5] Lack of discussion: As is often the danger when Results and Discussion sections are not separated, there is a lack of actual discussion. The discussion section should place the results in context of existing literature. This has not yet been done. Other evaluations of the models used exist. Other modelling studies have assessed different strategies for China or globally Etc. The authors need to place their results into such context, preferably by separating Results and Discussion into distinct sections.

We have reorganized section 4 to present our points more clearly. In the revised manuscript, the results and discussion are now divided into two distinct sections: "*Simulation and evaluation*" and "*Discussion*". The first section presents the simulation results, the materials we used to evaluate the model, and comparisons with these materials. The second section specifically addresses the differences among the models and explores the mechanisms behind these differences, aiming to enhance our understanding of the advances, limitations, and challenges in large-scale groundwater modeling.

Please also refer to the response to comment [2].

Minor Comments:

[6] Line 85ff.: The authors state that "Significant progresses or consensus have been achieved in community discussions regarding model parameterization, evaluation, calibration, and intercomparison". Given that at least the cited Gleeson et al. stresses the current lack of adequate evaluation strategies for global models, I would personally not frame it quite this positively. I do think that there is still significant advancement needed to derive at adequate strategies, and I also think that consensus is not yet there.

Corrected and cited new relevant papers, e.g., Reinicke et al. (ERL, 2024) and Devitt et al. (ERL, 2021). Please see lines 90-94 in the revised manuscript.

[7] Figure 6. The lower plots show positive and negative deviations from 0. The maps would be much clearer if the authors were to use a diverging color scheme as they do in Figure 7. Though I can also see that the authors prefer to keep the colors similar to the actual values.

Revised. They are in a new Figure 9 in the revised manuscript using a diverging color scheme.

**Researcher from the community:**

1) How did the authors handle uncertainty in datasets for potential recharge and soil properties in regions with sparse observational data, particularly in arid and semi-arid zones? Could more details on uncertainty quantification be provided?

Uncertainties of the large-scale hydrologic modeling are largely determined by uncertainties in the data products used. The generation of the input datasets is always a huge amount of work and separated from the modeling, i.e., the dataset generation and the large-scale hydrologic modeling are the focuses of two different communities and this division will be clearer moving forward in the big-data era. We mentioned that if it is five years earlier, such a modeling is impossible as many global data products were not available. As one of the most important efforts in our modeling, we tried to choose the best available datasets at current stage to reduce potential uncertainties. However, quantifying the intrinsic uncertainties in these data products are out of the scope of our work. Future work incorporating local available data is necessary to further improve the quality of the input datasets or decrease the uncertainties in them. One of our goals in this regard is to keep an eye on the advances in relevant data products in the community and dynamically replace some of the inputs with those of higher qualities at a feasible frequency.

Regarding the selection of datasets in our modeling, we have lengthy discussions for both potential recharge and soil properties. Please refer to lines 213 to 242 and lines

244 to 279 in the revised manuscript, respectively. We also briefly summarized them here as below.

As we mentioned in the manuscript, our objective is to continuously improve the workflow of large-scale surface water-groundwater modeling using ParFlow for community use globally. Therefore, we started from the workflow of CONUS 2.0. Then we found replacements of some datasets, e.g., those existing in US but are absent in China, or those having better ones in China area. For soil texture and deep geology, we used the same datasets GSDE and GLHYMPS 1.0. For flow barriers, there is a better data product for China area, so we replaced the global one by the new one. Also, all of them are the datasets well recognized by the community, i.e., the best choice we can use in China area not only because of CONUS 2.0 using them. Additionally, the combination of these datasets showed outstanding performance when they were tested in three large basins (the Upper Colorado River basin, the Little Washita basin, and the Delaware-Susquehanna Basin) based on ParFlow simulations evaluated by observed streamflow and water table depth.

For potential recharge (P-ET), we compared those generated by different precipitation and ET products and further constrained them with prior knowledge. We collected the latest P and ET products with higher spatial resolutions and long enough durations. Then we further filtered out those contrasting to prior knowledge. This is easy to do as it is well-known that P and ET products are of high uncertainties. For example, we know there is recharge in the upstream of Heihe River Basin, so the combinations of P and ET generating zero or negative potential recharge in this area will not be considered any more. In the manuscript, we also highlighted such significant uncertainties in the products challenging both the data and modeling communities. We also provided a possible solution in future work to generate P and ET products under a unified modeling framework constrained by the water balance.

2) The CONCN 1.0 model covers a vast area at high resolution, which demands substantial computational resources. Could the authors discuss any measures taken to

optimize computational efficiency and how the model's scalability could be extended to similar hydrologic regions?

Yes. We used seepage face as the top boundary condition in the first phase of the spinup and then turned on the overland flow in the second phase. This avoids the meaningless surface water-groundwater exchange in the early stage which mainly stabilizes the groundwater. For the scalability, we also have some experience. The CONCN model and the CONUS 2.0 model have very similar dimensions. Therefore, they take approximately the same wall clock time for spinup. Yet due to the larger area of arid and semi-arid regions in China, where the on and off of overland flow (integrated or groundwater only) may take more time to converge. Thus, the spinup of CONCN model takes slightly longer time. Additionally, ParFlow has excellent parallel scalability for different domain sizes and heterogeneities, which has been carefully tested and discussed in Ashby and Falgout (1996).

3) Would the authors consider using coarser resolution or data assimilation techniques to make the model more computationally accessible, particularly for policy-making applications?

Might be a choice but it is really hard to say that this is what we expect. Coarse resolutions will miss a lot subgrid variations which will cause the deviations of the simulation results. This is a well-known issue in the community of earth system modeling. A model with a higher resolution generally shows better performance if the parameterization is reliable enough (or similar). Thus, we are trying to build a high-resolution model to ensure the model performance instead of moving backward.

4) I recommend that the authors consider including a comparison with data assimilation approaches to enhance model accuracy and reduce uncertainties, especially in data-scarce regions. Data assimilation has been effectively applied in hydrologic modeling to integrate observed data with model predictions, often improving the alignment with real-world conditions. Techniques like Kalman filters or variational data assimilation could complement the current workflow, particularly for improving estimates of potential

recharge and water table depth in arid and semi-arid regions where observational data is limited. A comparison with data assimilation methods may also highlight the strengths of the CONCN model and provide a pathway for future enhancements in large-scale hydrologic modeling.

Data assimilation is an efficient approach for incorporating observations and doing parameter inversion. This is in our future plan of our modeling platform. The foundation of a sustainable modeling platform is to build a model of reliable/acceptable performances, i.e., the very first thing. Then the strength of the data assimilation can be fully leveraged. As a result, our first step focuses on the model structure, data selection, spinup, evaluation etc. We also identified the challenges in the modeling. It has been a huge step from scratch, and costs more than two years involving all authors and other collaborators. We are not to build a perfect modeling platform with everything in one paper which is impossible in a short time. We are doing step by step to gradually improve the modeling platform and timely share the results of each step with the community.

Once there is a model with acceptable performance, not only data assimilation but also many other approaches, e.g., emulators, could be incorporated into this modeling framework. The data assimilation improves some of the parameters relying on the observations of some others. This means it still has high requirements of observations. As mentioned by reviewer, it is data scarce in arid and semi-arid areas. Collecting long-term observations of enough spatial density, e.g., water table depth, which has been confirmed useful in data assimilation of groundwater modeling, is a big challenge. In some regions, it is even impossible as the observation network has not been built. Therefore, we also discussed in the manuscript that, moving forward, this modeling platform needs collaborative efforts from different communities.

5) To strengthen the contextual foundation of this study, I recommend the authors cite established integrated hydrologic models like SWAT-MODFLOW in the introduction. SWAT-MODFLOW, widely used for its integration of surface and subsurface processes, has significantly advanced our understanding of coupled surface-groundwater systems

across various scales. Citing SWAT-MODFLOW alongside ParFlow and other large-scale models would provide readers with a broader perspective on the tools available for integrated hydrologic modeling. This comparison may also underscore the unique challenges and innovations of applying ParFlow within China's hydrologic and geologic context, while highlighting the importance of diverse model approaches for managing complex water resources.

I strongly recommend to cite below paper:

"Assessing regional-scale spatio-temporal patterns of groundwater–surface water interactions using a coupled SWAT-MODFLOW"

"Assimilation of sentinel-based leaf area index for modeling surface-ground water interactions in irrigation districts"

"Development and application of the integrated SWAT–MODFLOW model."

Our objective is in the framework of large-scale hydrologic modeling. We have done substantial literature review and listed the latest large-scale hydrologic models either in China or at global/national scale, including those using MODFLOW. Though SWAT-MODFLOW is relevant to integrated hydrologic modeling, these three papers are neither relevant to China nor to global/national scale. We fully respect reviewer's strong desire to cite the new published WRR paper, so we cite all three papers in the discussion. Please refer to line 596 in the revised manuscript.

Overall, we appreciate reviewer's interesting thoughts and are pleasant to exchange our ideas on these thoughts. However, these thoughts are more or less deviated from the objective of this work or beyond the scope of this very first and important step. Open questions remain in the large-scale high resolution groundwater modeling. As we mentioned in the discussion, all three groundwater models show different water table depths implying large uncertainties. This is more challenging in an integrated framework and in a data-poor region. Therefore, we are trying to use what we can use in such a region to build a model with acceptable performances (actually, it is unexpected

excellent performances) as a reference for the community. The data/datasets collection, selection, processing, assembling the model, fetching computational resources, running the model, and analyzing the simulation results and comparing them with previous models, etc., have been substantial work. We don't aim to finish everything in one step, but to gradually improve it with time.

**References in this letter:**

Fan, Y., Li, H., and Miguez-Macho, G.: Global Patterns of Groundwater Table Depth, Science, 339, 940-943, 10.1126/science.1229881, 2013.

Fan, Y., Miguez-Macho, G., Jobbágy, E. G., Jackson, R. B., and Otero-Casal, C.: Hydrologic regulation of plant rooting depth, Proceedings of the National Academy of Sciences, 114, 10572-10577, 10.1073/pnas.1712381114, 2017.

Fan, Y., Miguez-Macho, G., Weaver, C. P., Walko, R., and Robock, A.: Incorporating water table dynamics in climate modeling: 1. Water table observations and equilibrium water table simulations, Journal of Geophysical Research: Atmospheres, 112, https://doi.org/10.1029/2006JD008111, 2007.

---

## Author Response (AR2)

Dear Authors,

Your article is close to final approval. However, as you can see from the comments of the two reviewers, some minor changes have been requested.

Please consider these suggestions and submit a new revision along with a document showing your changes.

Thanks for the further comments from the editor and two reviewers. We addressed all comments in this revision. Please see the response below. All line numbers refer to that in the changes-tracked manuscript. We also made minor editions in other places to improve the readability.

I just have a few details:
- p 6, l.173: Correct: Indicators

Corrected, please see line 149.
- p 19, l.536: Correct "decease" to "decrease"?

Corrected, please see line 455.
- p 21, l.674: "... we did not ...".

Corrected, please see line 475.
- p 22, l.813: "use the difference"

Corrected, please see line 494.
- p 24, l.875: "to the global models"

Corrected, please see line 555.

[1] Abstract: You do not define RSR in the abstract, which makes it impossible to understand the abstract before reading the paper. The same is true for the introduction section of the paper.

We added the definition of RSR into the abstract and the introduction. Please see lines 34–35 and 129–130.

[2] Line 154: "An RSR value of 1.0 suggests a good performance while 0.5 suggests an excellent performance (O'neill et al., 2021)." While I appreciate the additional comments made by the authors on this point, I still think this sentence is misleading and should be taken out. The authors use the same statements of excellent performance etc later in the manuscript. It is sufficient to state what the optimal value is given that the rest is purely subjective opinion. Currently you explain that this subjective analysis should not be done, but then you do it.

In this revision, we edited this sentence to "In this study, we expect a satisfied performance with an RSR value less than 1.0 while 0.5 is preferable". We also removed all "good" and "excellent" anywhere in the text and figures. Please see lines 35, 37, 130–131, 211, 213, 389, 431, 553, and new figures 2 and 4.

[3] Figure 10: I like the addition of this figure which is interestingly different to the previous study

mentioned. The additional discussion is now placing the results in the context beyond the models directly used in this study, as does the added discussion of Karst.

Thanks!

[4] The Summary and Outlook section is appropriately titled. One additional consideration to point (4) could be that much useful information is probably hidden in system conceptualizations done for previous modelling studies. Particularly in the context of groundwater system, hydro-geologists have interpreted available data in many places before – which holds valuable information due to the regional combination of expertise and data.

Good point. We added this suggestion into the manuscript. Please see lines 608–613.

---

## Author Response (AR3)

The ROR database lists the institution of the corresponding author but with a different city than given in the manuscript. Please clarify whether the ROR "Sun Yat-sen University (Guangzhou, China)" is still correct.

(Guangzhou, China) is wrong. It should be (Zhuhai, China). I cannot find (Zhuhai, China) in ROR, so I used (Guangzhou, China).